# Dynamic CpG methylation delineates subregions within super-enhancers selectively decommissioned at the exit from naive pluripotency

Emma Bell [1,6], Edward W. Curry[1,6], Wout Megchelenbrink[2,3,4,6], Luc Jouneau[5], Vincent Brochard[5], Rute A. Tomaz [1], King Hang T. Mau[1], Yaser Atlasi[2], Roshni A. de Souza[1], Hendrik Marks [2], Hendrik G. Stunnenberg [2,3], Alice Jouneau[5,7✉] & Véronique Azuara [1,7✉]

Clusters of enhancers, referred as to super-enhancers (SEs), control the expression of cell identity genes. The organisation of these clusters, and how they are remodelled upon developmental transitions remain poorly understood. Here, we report the existence of two types of enhancer units within SEs typified by distinctive CpG methylation dynamics in embryonic stem cells (ESCs). We find that these units are either prone for decommissioning or remain constitutively active in epiblast stem cells (EpiSCs), as further established in the peri-implantation epiblast in vivo. Mechanistically, we show a pivotal role for ESRRB in regulating the activity of ESC-specific enhancer units and propose that the developmentally regulated silencing of ESRRB triggers the selective inactivation of these units within SEs. Our study provides insights into the molecular events that follow the loss of ESRRB binding, and offers a mechanism by which the naive pluripotency transcriptional programme can be partially reset upon embryo implantation.

[1] Institute of Reproductive and Developmental Biology, Faculty of Medicine, Imperial College London, Hammersmith Hospital, Du Cane Road, London W12 0NN, UK. [2] Radboud University, Faculty of Science, Department of Molecular Biology, 6525GA Nijmegen, the Netherlands. [3] Princess Máxima Center for Pediatric Oncology, Heidelberglaan 25, 3584 CS Utrechtt, the Netherlands. [4] Department of Precision Medicine, University of Campania Luigi Vanvitelli, Vico L. De Crecchio 7, 80138 Napoli, Italy. [5] Université Paris-Saclay, INRAE, ENVA, BREED, Domaine de Vilvert, bat 230, 78350 Jouy-en-Josas, France. [6] These authors contributed equally: Emma Bell, Edward W. Curry, Wout Megchelenbrink. [7] These authors jointly supervised: Alice Jouneau, Véronique Azuara. ✉email: alice.jouneau@inrae.fr; v.azuara@imperial.ac.uk

Pluripotency, the ability to form all tissues in an adult organism, is under the control of complex mechanisms that enable cells to differentiate into the early somatic and germ cell lineages. Embryonic stem cells (ESCs) and epiblast stem cells (EpiSCs) are derived from mouse pre- and post-implantation embryos, respectively, and represent the naive and primed state of pluripotency[1]. Alongside the core transcription factors (TFs) OCT4, SOX2 and NANOG (OSN), ESC identity is associated with the expression of an additional cohort of naive pluripotency factors, including ESRRB, KLF4, TBX3 and TFCP2L1. These proteins are highly expressed in ESCs and downregulated as primed pluripotency is established[2]. Conversely, OTX2, ZIC2 and OCT6 factors were identified as major transcriptional regulators of primed EpiSCs in the context of the continued expression of OSN[3–5].

Enhancers act as hubs of TF binding and promote gene expression. Recent studies suggest that large regions with clustered enhancer units, often described as super-enhancers (SEs), regulate the expression of key cell identity genes[6–8]. Readily demarcated by enhancer-specific histone marking and protein binding at high density, SEs in ESCs preferentially recruit numerous naive TFs, and are predicted to be decommissioned as cells exit from naive pluripotency[8]. Given the shared expression of a large panel of genes in ESCs and EpiSCs[9,10], it remains unclear how this is achieved. In particular, the fate of individual enhancer units across SEs has not been investigated during the transition from naive-to-primed pluripotency.

In this study, we identify molecular and functional differences between enhancer units within SEs, revealing distinctive regulatory mechanisms. We find that enhancer units mapped in ESCs divide into two types based on whether or not they continue to function in EpiSCs, as further established in the post-implantation epiblast in vivo. Mechanistically, we demonstrate that ESC-specific enhancer units exhibit extensive cell-to-cell CpG methylation heterogeneity and are most specifically marked by ESRRB. As a result, these units are selectively destabilised at the exit from naive pluripotency, as recapitulated upon ESRRB depletion. Loss of ESRRB in ESCs promotes de novo methylation, reduces mediator and RNA polymerase II (POL2) binding and attenuates the expression of target genes and enhancer RNAs (eRNAs) over disruption of chromatin interactions. In contrast, ESRRB-independent units within SEs remain active and hypomethylated through steady binding of TFs and co-regulators in ESCs and EpiSCs. These units promote the expression of a core set of genes throughout the naive-to-primed transition, suggesting a crucial role for the upholding of pluripotency upon embryo implantation.

## Results

### Hypomethylation delineates active SE units.
SEs were defined in ESCs based on high-density binding of the mediator component MED1 and deposition of H3K27ac histone marks over large genomic regions in contrast to typical-enhancers (TEs)[8]. Additionally, we noticed that SEs and TEs show a significant difference in GC content (Supplementary Fig. 1a). While the median CpG density in TEs approximates the mouse genome average, SEs present higher GC content and CpG density (Wilcoxon rank-sum test, $p < 2 \times 10^{-16}$). Thus, we hypothesised that CpG methylation might contribute to the structural organisation of SEs. To test this idea, we used available bisulfite-sequencing (BS-seq) data collected from ESCs grown in the presence of serum and leukemia inhibitory factor (serum/LIF)[11]. We scored each CpG as methylated (mCpG) or unmethylated along SEs mapped in ESCs and other cell types[8] using a Hidden Markov model (HMM; see Methods section). As anticipated, high CpG methylation levels

were steadily detected at somatic (proB cell) SEs (Supplementary Fig. 1b), in keeping with their inactive status in pluripotent cells. In contrast, ESC SEs displayed a complex profile consisting of low and intermediate levels of CpG methylation. Interestingly, ProB cell SEs showed a similar low-to-intermediate profile in haematopoietic cells[12] (Supplementary Fig. 1c), highlighting the close relationship between CpG methylation and cell identity[13,14]. Further inspection of individual ESC SEs identified discrete unmethylated subregions in ESCs, which overlapped with H3K27ac deposition and binding of pluripotency TFs and co-regulators, as depicted for the *Klf4*-associated SE (Fig. 1a). By computing protein binding (chromatin immunoprecipitation sequencing (ChIP-seq)) and chromatin accessibility (ATAC-seq) at unmethylated and methylated subregions across all SEs using published datasets (Supplementary Data 1), we validated that local hypomethylation demarcates active SE units in ESCs (Fig. 1b).

To verify whether promoter–SE interactions preferentially establish within unmethylated over methylated regions, we studied the chromatin interactions between these subregions and target gene promoters at high resolution in ESCs. For this, we interrogated available (promoter) capture Hi–C libraries generated using 4 bp recognition restriction enzymes[15,16] and called significant promoter–SE interactions with the CHICAGO pipeline[17] (Fig. 1c; Supplementary Fig. 2 for additional examples). Given the high correlation ($R = 0.77$) between the selected datasets (Supplementary Fig. 3a, b), all significant promoter–SE interactions identified from either Joshi et al.[15] or Sahlen et al.[16] studies (629 in total) were considered (Supplementary Data 2). These included previously described SE-interacting promoters using cohesin CHIA-PET[18] and an alternative promoter capture Hi–C based on a 6 bp recognition restriction enzyme[19] (Supplementary Fig. 3c, d), and were significantly enriched in Gene Ontology terms related to embryonic development as expected (Supplementary Fig. 3e). Using this method, we demonstrated that unmethylated subregions engaged more frequently with active promoters than methylated subregions within SEs (Fig. 1d, Supplementary Fig. 3h). Collectively, our findings confirm that SEs are intrinsically heterogeneous, consisting of one or more hypomethylated active enhancer units in ESCs.

### Differential inactivation of SE units in EpiSCs.
The transition from pre- to post-implantation is marked by a global increase in CpG methylation as reported in vitro and in vivo[20–22]. Using newly generated BS-seq data in EpiSCs (this study), we observed that SEs mapped in ESCs accumulate substantial levels of CpG methylation in the primed cells (Supplementary Fig. 4a). By contrast, ESC SE-interacting promoters remained largely hypomethylated (Supplementary Fig. 4a–c). Strikingly, however, we found that CpG methylation was acquired in different patterns across individual ESC SEs (Fig. 2a). While all enhancer units of some SEs were targeted by high levels of CpG methylation (e.g., *Klf4*-associated SE), specific units escaped methylation in other SEs (e.g., *Klf13*- and *Lefty1*-associated SEs). By comparing the methylation status of all SE units in ESCs and EpiSCs, we thus identified two types of units with different fates: persistently unmethylated (PU, green) and differentially methylated (DM, magenta; Fig. 2b, Supplementary Fig. 4d). Outside these regions, CpG methylation was consolidated from ESCs to EpiSCs (interstitial regions; INT, grey), largely contributing to the hypermethylated profile of SEs as a whole in EpiSCs (Supplementary Fig. 4a).

To validate the assignment of PU and DM subregions in independent ESC and EpiSC lines, we probed alternative BS-seq studies[14,23]. To test whether the remodelling of SEs occurs in a

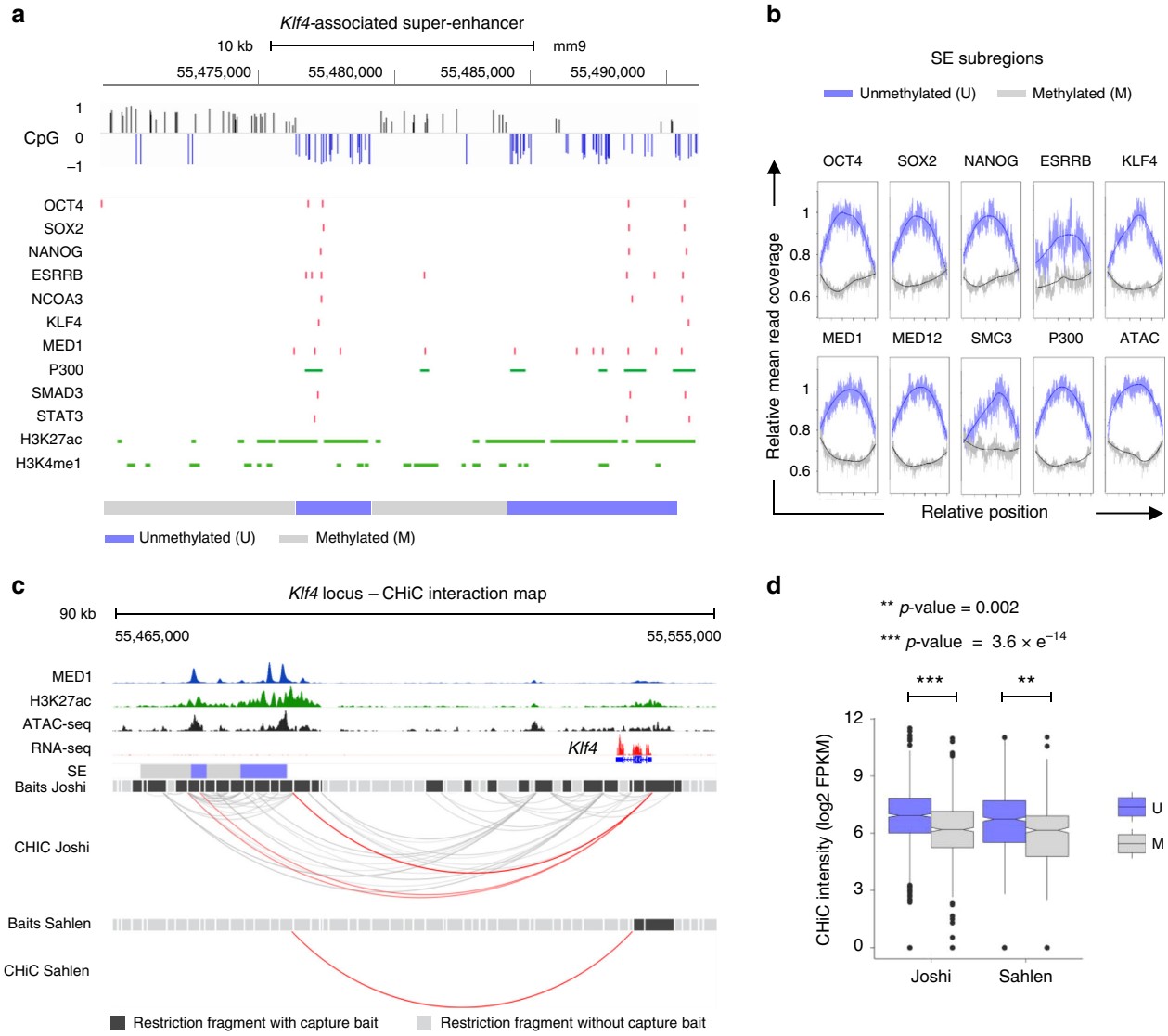

**Fig. 1 Demarcation of active enhancer units within SEs through CpG methylation profiling in ESCs. a** Schematic representation of the *Klf4*-associated SE in embryonic stem cells (ESCs) grown in serum/LIF, showing CpG methylation, TF binding sites and epigenetic marks along the SE locus. The methylation state of each CpG was determined using a HMM: unmethylated = blue, down; methylated = grey, up. In addition, the height of the bars indicates the extend of methylation (positive values) and demethylation (negative values) of each CpG (see also Methods section). Coordinates of the *Klf4*-associated SE are shown as reported in ref. [8]. **b** Mean Chip-seq read coverage of TFs, mediator and cohesin subunits, P300 and ATAC-seq signals at unmethylated (U; blue) and methylated (M; grey) SE subregions in ESCs (serum/LIF). The mean read coverage scores were scaled to the maximal value for either region set. Publicly available datasets used are listed in Supplementary Data 1. **c** Promoter–SE interactions along the *Klf4* locus. The capture baits (black) for the Joshi et al.[15] and Sahlen et al.[16] capture Hi–C interactions are shown. Key epigenetic enhancer marks (MED1, H3K27ac and ATAC-seq) overlap with unmethylated SE subregions (U; blue) across *Klf4*-associated SE, while methylated regions (M; grey) are largely devoid of active histone marks. Promoter–SE and intra-SE interactions are shown as red arcs; other interactions as grey arcs. **d** Box plots of the capture Hi–C interaction intensity (log2 fragments per kilobase million (FPKM)) of unmethylated (U; blue) and methylated (M; grey) SE subregions for which the SE has a significant interaction with the target promoter. For this analysis, only expressed SE-interacting gene promoters were considered (reads per kilobase million (RPKM) ≥ 1). Unmethylated SE subregions interact at a significantly higher frequency as compared to methylated SE subregions with an equal number of capture baits (**$p = 0.002$; ***$p = 3.6 \times 10^{-14}$, analysis of variance, controlling for the number of capture Hi–C baits per SE subregion and the log2 promoter–SE subregion distance).

step-wise manner, we also examined the profile of ESC-derived epiblast-like cells (EpiLCs)[23], offering a transitional stage between naive and primed identities. In agreement with our data, PU compared to DM subregions appeared largely unmethylated in EpiLCs and EpiSCs (Fig. 2c, Supplementary Fig. 4e). DM subregions in EpiLCs adopted an intermediate level relative to PU and INT regions, becoming highly methylated in EpiSCs. To further determine whether PU and DM subregions were also differentially methylated in the developing embryo, we

processed available BS-seq data in ICM/epiblast tissues dissected from pre- (E3.5 and E4.0) and post-implantation (E5.5 and E6.5) embryos[22]. Importantly, a similar pattern of CpG methylation was recapitulated in vivo, with DM subregions becoming gradually and selectively decommissioned. During this process, differential CpG methylation at PU, DM and INT was already apparent starting from E3.5–E4 (Fig. 2d), which coincides with the onset of de novo CpG methyltransferases expression in the developing epiblast[24]. Collectively, our findings reveal that

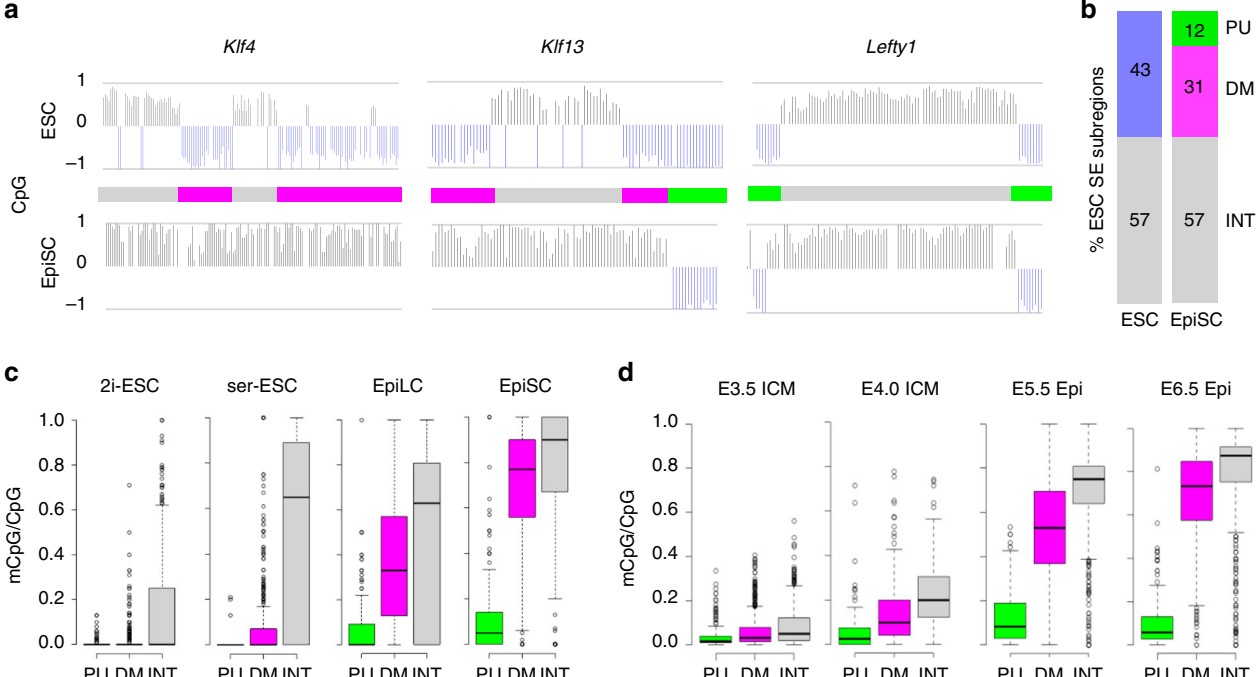

**Fig. 2 ESC SEs acquired CpG methylation in different patterns as primed pluripotency is established. a** CpG methylation along three representative SEs in embryonic stem cells (ESCs; serum/LIF)[11] and epiblast stem cells (EpiSCs; this study) in CDM supplemented with activin A and fibroblast growth factor 2 as represented in Fig. 1a. Unmethylated SE subregions that gain methylation in EpiSCs are represented as magenta bars (DM subregions), and those that remain unmethylated as green bars (PU subregions). Grey bars represent interstitial (INT) regions in ESCS that remain methylated in EpiSCs. **b** Proportion (%) of SE subregions in ESCs and EpiSCs. Unmethylated regions in ESCs (blue bar) either remain unmethylated (PU, green, 184 subregions) or become methylated (DM, magenta, 492 subregions) in EpiSCs; INT (547 subregions) methylated regions are shown in grey. A total of 231 SEs were analysed. **c** CpG methylation levels (mCpG/CpG) at PU, DM and INT subregions in an independent ESC line grown in serum/LIF (ser)[14], in EpiLCs and an independent EpiSC line[23]. ESCs grown in 2i/LIF (2i)[11] are also shown; 2i/LIF conditions enforce a globally hypomethylated state of genome. **d** CpG methylation levels (mCpG/CpG) at PU, DM and INT subregions in vivo in pre-implantation (inner cell mass (ICM); E3.5 and E4.0) and post-implantation epiblast (Epi; E5.5 and E6.5) tissues[22].

enhancer units within SEs partition as constitutively hypomethylated (PU) or decommissioned (DM) during the pre- to post-implantation transition.

**DM and PU units regulate distinct pluripotency gene modules.** As a functional readout of differential CpG methylation within SEs, we tested whether the expression of SE-interacting promoters was altered in vitro (from ESCs to EpiSCs) and in vivo (from ICM E3.5 to epiblast E6.5)[22,25]. Expression changes were predicted to be inversely related to average CpG methylation levels and compared with expression dynamics measured by RNA-seq (see Methods section). Amongst most affected SE-interacting promoters, we identified well-established naive pluripotency factors, including *Klf4*, *Esrrb*, *Prdm14*, *Tbx3*, *Tcfcp2l1* and *Zfp42* (Fig. 3a; magenta dots). These silenced promoters were associated with SEs that only contain DM subregions (Supplementary Data 3). In contrast, the expression of other genes was predicted to be less affected, including *Klf13*, *Lefty1*, *Med13l*, *Otx2*, *Pou5f1*, *Nanog*, *Sox2* and *Tet1* (Fig. 3a; green dots). These promoters were expressed or even upregulated in the primed cells and associated with SEs that contain at least one PU subregion.

To follow-up on this observation, we divided all ESC SEs into two classes: class I SEs that enclose at least one PU subregion, and class II SEs that only contain DM subregions (Fig. 3b), and asked whether the two classes of SEs regulate distinct gene repertoires. Given that multiple SEs can interact with the same active gene promoters though at variable frequency (Supplementary Data 2), we focused on the closest interacting genes, which indeed correlate with the strongest SE–promoter interactions (Supplementary

Fig. 3f, g). Class I (light green) and class II (pink) SE-associated genes showed comparable expression levels in ESCs (Supplementary Fig. 4f). Correlating with the presence of PU units, class I genes overall remained expressed in EpiSCs. In contrast, class II genes showed a significant trend for downregulation relative to ESCs (Kruskal–Wallis test, $p = 4 \times 10^{-7}$). Importantly, these contrasting gene expression dynamics were recapitulated in vivo from E3.5 to E6.5 (Fig. 3c; $p = 5.5 \times 10^{-3}$), mirroring the segregation of PU (unmethylated) and DM (methylated) subregions in the peri-implantation epiblast (Fig. 2d).

To determine whether the differential transcriptional fates of class I and class II SE-associated genes diverge in tandem at the exit from naive pluripotency, we performed reverse transcription-quantitative PCR (RT-qPCR) analysis of selected candidates at different time points upon the conversion of ESCs into EpiSCs in vitro[26,27] (Fig. 3d, Supplementary Fig. 5a–c). As anticipated, we found that class II SE-associated genes analysed consistently loose expression starting from day 1 post induction. In contrast, class I candidates remained largely expressed through conversion and upon acquisition of primed pluripotency. Collectively, our results suggest that SEs containing DM subregions only (class II) regulate the expression of genes associated with the naive ESC state. In contrast, SEs containing at least one PU subregion (class I) remain active in primed EpiSCs, promoting the expression of a core set of pluripotency genes throughout the naive-to-primed cell state transition. In agreement with this conclusion, DM subregions showed declining H3K27ac deposition, OCT4 binding and accessibility (ATAC-seq) from ESCs to EpiSCs (Fig. 3e), indicative of

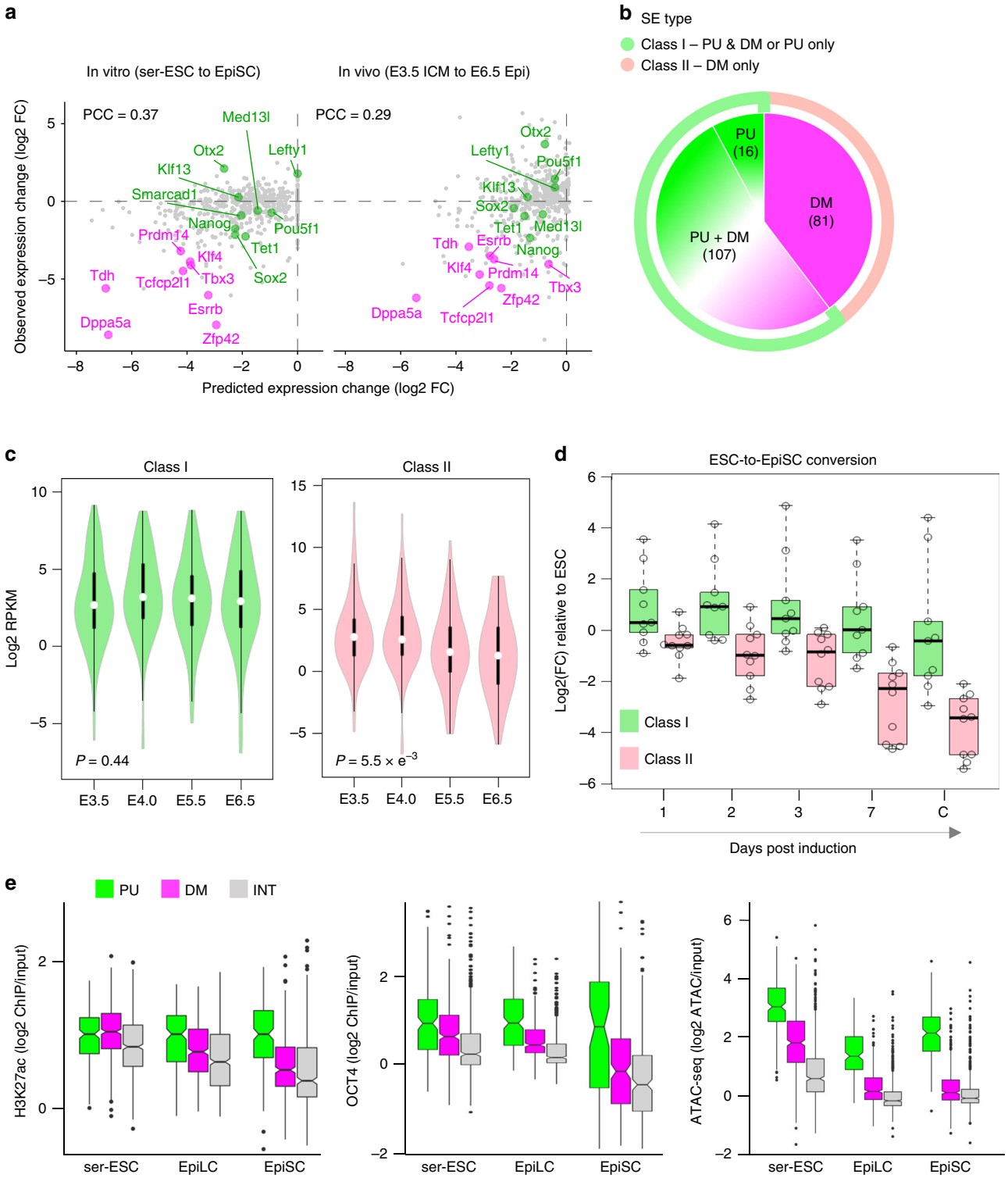

decommissioning as previously described at naive enhancers[3]. In contrast, PU subregions retained an active enhancer status, and maintained OCT4 and SOX2 binding along with primed pluripotency TFs (OTX2 and ZIC2)[4,5] in EpiLCs and EpiSCs (Fig. 3e, Supplementary Fig. 5d, e), indicative of continued activity.

**Cell-to-cell methylation heterogeneity at DM in ESCs.** ESCs in serum/LIF can toggle from naive-to-primed pluripotency states

with some cells initiating differentiation, as reflected in heterogeneous transcriptional states[28,29]. Interestingly, cell state fluctuations also manifest at the epigenetic level with evidence of CpG methylation oscillations at enhancers[30–33]. To assess the level of cell-to-cell methylation heterogeneity at PU and DM, and its potential association with gene expression, we reanalysed parallel single-cell transcriptional (sc-RNA-seq) and bisulphite (sc-BS-seq) data from ESCs grown in serum/LIF and 2i/LIF[30]. Remarkably, substantial variation in CpG methylation levels was

**Fig. 3 Partitioning of enhancer units within SEs as constitutively active or decommissioned in primed cells. a** Predicted and observed gene expression fold change (log2) of SE target genes in vitro (left) and in vivo (right). The predicted fold change is the negative product of the interaction intensity (log2 normalised reads) in ESCs (serum/LIF; ser) and CpG methylation changes (percent change) at PU and DM subregions from ESCs to EpiSCs (in vitro) or E3.5 ICM to E6.5 Epi (in vivo; see Methods section and Supplementary Data 1). Well known pluripotency genes associated with different categories of SE subregions are highlighted in magenta (DM) and in green (PU). PCC, Pearson's correlation coefficient. **b** Venn diagram depicting SE subregion composition and number of SEs falling into class I (light green) and class II (pink) categories. Class I SEs contain PU or PU + DM subregions, while class II SEs only contain DM subregions. **c** Expression (log2 RPKM) of all closest interacting gene promoters associated with class I or class II SEs in epiblast cells in vivo at different developmental times (from E3.5 to E6.5)[22]. p-Values (Kruskal–Wallis test) are indicated. **d** Expression changes (log2(fold change)) relative to ESCs (serum/LIF) of selected genes associated with class I (light green) or class II (pink) SEs during the conversion of ESCs into EpiSCs in vitro (RT-qPCR). Class I candidates are: *Lefty1, Pou5f1, Otx2, Smarcad1, Sox2, Tet1, Klf13, Med13l* and *Nanog*; class II candidates are: *Esrrb, Klf4, Prdm14, Tbx3, Tdh, Tet2, Tfcp2l1, Klf2, Klf5* and *Zfp42*. Statistical analysis was performed using non parametric tests for repeated measures data in factorial designs. The factorial design chosen corresponds to f1.ld.f1 in nparLD R-package. Kinetics of the two classes were statistically different ($p = 4.5 \times 10^{-12}$, Wald test). Data represent three independent conversion experiments ($n = 3$). Source data are provided as Supplementary Data 10. **e** H3K27ac deposition, OCT4 binding and chromatin accessibility (ATAC-seq) of PU, DM and INT subregions in ESCs (serum/LIF; ser), EpiLCs and EpiSCs. Publicly available datasets used are listed in Supplementary Data 1.

revealed at DM subregions in serum/LIF conditions (Fig. 4a). In contrast, PU subregions were stably hypomethylated in all cells examined. As expected, heterogeneity across SE subregions was less apparent in 2i/LIF conditions, which promote global genome hypomethylation in ESCs[11,24].

Using hierarchical clustering (see Methods section), we identified two subpopulations in serum/LIF ESCs: "naive-like" cells showing hypomethylated DM subregions as seen in 2i/LIF, and "primed-like" cells harbouring higher methylation level and variance at the same regions (Fig. 4a, b, Supplementary Fig. 6a). CpG methylation dynamics at DM subregions were also evident when comparing the profiles of individual class I (*Med13l*) and class II (*Esrrb*) SEs in the two cell clusters (Fig. 4c and Supplementary Fig. 6b for additional examples). These modulations correlated with changes in the expression of class II but not class I SE-associated genes, suggesting functional importance. Here, class II genes were highly expressed in "naive-like" cells only (Supplementary Fig. 6c, d), in agreement with the DM hypomethylated status of these cells. This suggests that epigenetic heterogeneity at DM subregions might selectively destabilise the expression of ESC-specific genes, enabling their acute down-regulation upon exit from naive pluripotency.

Given the importance of PU subregions, we sought to investigate how these enhancer units are protected from similar CpG methylation dynamics in ESCs. Interrogating available bulk ESC ChIP-seq datasets in serum/LIF (Supplementary Data 1), we found that PU relative to DM and INT subregions harboured higher enrichment for H3K4me3 in contrast to H3K4me1 or H3K27ac (Fig. 4d, left panel). H3K4me3 is known to repel the binding of de novo DNA methyltransferases DNMT3A and DNMT3B (DNMT3s), possibly leading to less CpG methylation at these sites[34,35]. We therefore compared the relative occupancy of DNMT3s at SE subregions, along with the antagonistic enzyme TET1, which is capable of removing CpG methylation in a multistep process[36]. As anticipated, PU subregions showed significantly lower DNMT3s occupancy and higher recruitment of TET1 (Fig. 4d, right panel). By comparison, DM subregions appeared to be co-bound by TET1 and DNMT3s, especially DNMT3A known to target naive enhancers upon ESC differentiation[37]. To evaluate whether PU and DM subregions could be predicted based on these epigenetic signatures, logistic regression models were fitted with subregion type as a binary outcome (DM vs PU) and a set of individual features as quantitative predictor variables (see Methods section). All features tested apart from H3K27ac were significantly associated with DM/PU status (Supplementary Data 8). Increased ChIP-seq enrichment for DNMT3s and H3K4me1 were found highly predictive of the DM status, while TET1, H3K4me3 and ATAC-seq signals were most closely associated with the PU status. CpG density also appeared

to be a better predictor of PU units, in coherence with TET1 preferential binding to CpG-rich regions[38–40]. Interestingly, 17% of PU subregions harboured high CpG density (above 0.05; Supplementary Fig. 6e), alone accounting for their hypomethylated status[41] as predicted.

To further our understanding of how DNMT3s and TET binding impacts on CpG methylation dynamics at the single-cell level, we used available sc-BS-seq data collected from DNMT3A/B double and TET1-3 triple knockout (KO) ESCs grown in serum/LIF[31]. Variance in CpG methylation at PU subregions remained mostly unchanged upon the loss of either DNMT3 or TET proteins (Fig. 4e), pointing to a non-exclusive protective role for TET binding at these subregions. In contrast, methylation variance at DM regions was highly reduced upon loss of DNMT3s and to a much lesser extend in TETs KO ESCs. This indicates that CpG methylation dynamics at DM subregions depend on the activity of de novo methyltransferases, and furthermore suggests that TET-mediated demethylation might not be a main driver of epigenetic heterogeneity at SEs, as similarly reported using allele-specific reporters of candidate SEs[33].

**ESRRB most specifically demarcates DM subregions within SEs.** To explore the additional regulators of CpG methylation at DM subregions besides DNMT3s, we used available sc-RNA-seq[30] to ask whether sporadic induction of early differentiation/primed pluripotency genes in serum/LIF[42] could play a part in the observed cell-to-cell epigenetic heterogeneity. Receiver operating characteristic (ROC) curves were generated to evaluate, on the basis of their normalised expression levels, the ability of co-expressed or individual genes to separate "naive-like" from "primed-like" single-cell clusters defined in Fig. 4a (see Methods section and ref. [43]). Notably, three gene sets were tested encoding for naive, general or primed pluripotency markers (Supplementary Data 4). Our results ruled out that the latter drives (as genetic oscillators) the metastable epigenetic state of DM subregions, instead pointing to a role for naive pluripotency factors (Fig. 4f, left panel). Among these factors, *Esrrb, Klf2* and *Rex1/Zfp42* (area under the ROC curve (AUC) values > 0.92, $p = 10^{-5}$) were identified as top genes whose increased expression best discriminates "naive-like" cells from "primed-like" cells (Fig. 4f, right panel; Supplementary Fig. 6f). In contrast, *Oct4, Klf4, Nanog* or *Nr5a2* provided less predictive power. These findings raise the possibility that heterogeneous expression and/or binding of specific naive pluripotency TFs could regulate the local CpG methylation dynamics and accessibility of SE subregions.

To interrogate the role of TF binding in defining distinctive chromatin states along SEs, we analysed TF motif enrichment in PU, DM and INT subregions, and examined the expression status

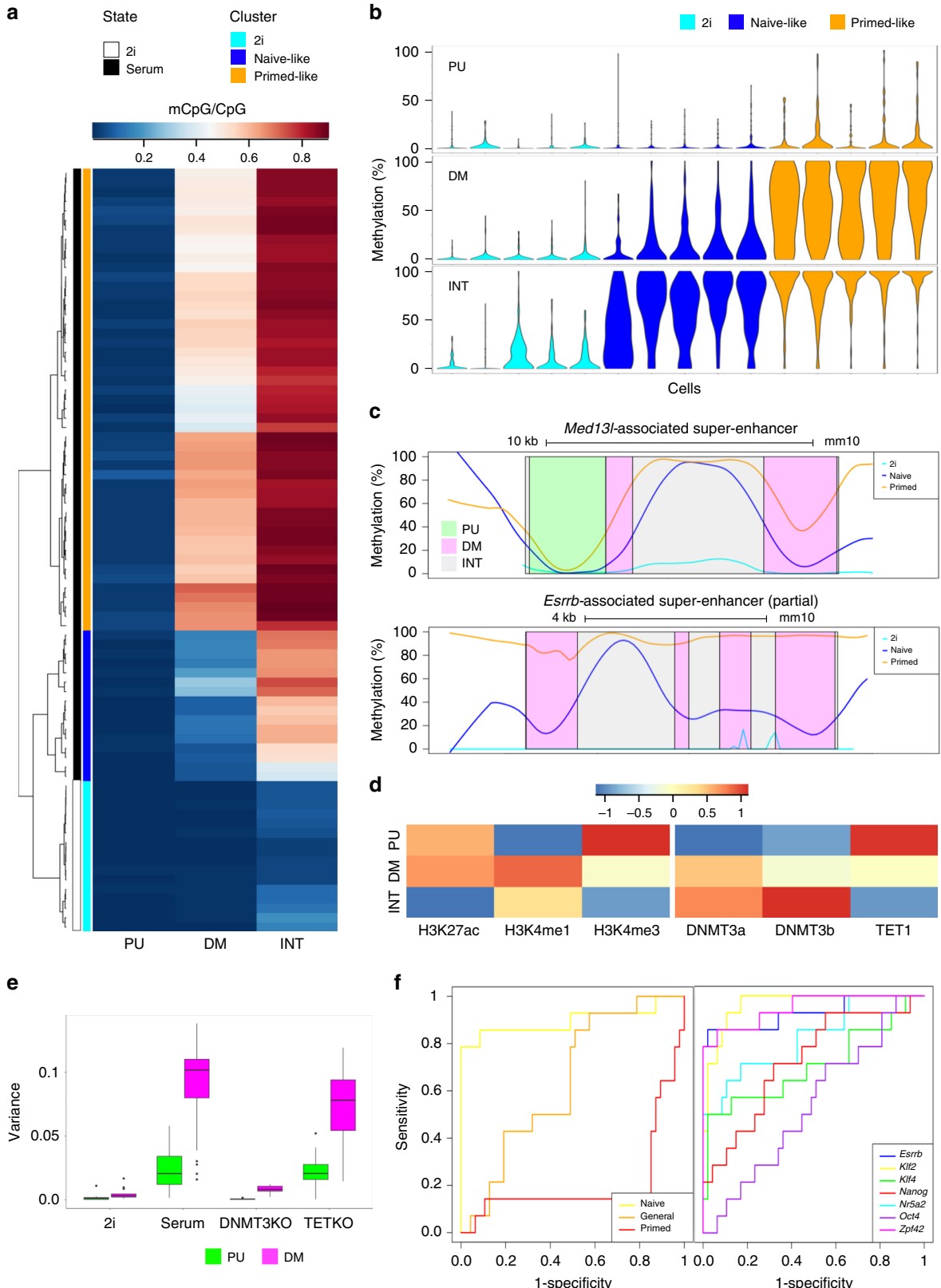

of TFs corresponding to these motifs (see Methods section). Approximately half of the statistically enriched motifs (hypergeometric test, Benjamini–Hochberg (BH) adjusted $p < 0.05$) in either PU or DM subregions were attributed to at least one corresponding TF expressed in serum/LIF ESCs (RPKM ≥ 1; Supplementary Fig. 7a). In contrast, INT subregions harboured

statistically enriched motifs that correspond to less frequently expressed TFs ($p < 3 \times 10^{-6}$) including differentiation-associated TFs induced later in development (Supplementary Data 5).

Interestingly, the cognate motif for ESRRB/NR5A2 showed strong enrichment in DM subregions only ($p < 5 \times 10^{-9}$), while the KLF/SP motif was found enriched in both DM ($p < 3 \times 10^{-7}$)

**Fig. 4 Contrasting CpG methylation dynamics at PU and DM subregions in individual ESCs. a** CpG methylation levels (mCPG/CpG) at PU, DM and INT SE subregions in 16 individual ESCs grown in 2i/LIF (2i, white) and 64 ESCs grown in serum/LIF[30] (serum, black). Each row represents one single cell, with blue–red colour gradient indicating proportion of methylated CpGs in the corresponding SE subregion class. Using hierarchical clustering, three cell clusters are identified: 2i (light blue), "naïve-like" (blue) and "primed-like" (orange) serum-ESCs. **b** Violin plots representing the distribution of the average methylation (%) across SE subregions for five randomly selected single cells in each of the three single-cell clusters defined in **a**. **c** Profiles of CpG methylation levels throughout the *Med13l-* (class I) and *Esrrb*-associated (class II) SEs with indication of PU (light green), DM (pink) and INT (grey) subregions. Each line gives the lowest-smoothed average of each individual CpG methylation across all individual cells in the corresponding cluster: 2i cluster (light blue), "naïve-like" cluster (blue) and "primed-like" cluster (orange). **d** Heatmaps showing the relative levels of enrichment (median log$_2$ fold change of ChIP-seq signal (RPKM + 1) over input signal (RPKM + 1)) for selected histone marks (left panel) and CpG methylation regulators (right panel) at PU, DM and INT subregions in bulk ESCs grown in serum/LIF. Enrichment scores for each feature are scaled by dividing by the mean enrichment score for that feature across all subregions. See Supplementary Data 1 for dataset accession numbers. **e** Box plots representing the distribution of methylation variances across PU (green) and DM (magenta) subregions for each individual cell analysed in different treatment conditions (2i and serum wild-type ESCs, and *Dnmt3a/b* DKO and *Tet1-3* TKO ESCs grown in serum/LIF)[30,31]. **f** Receiver operating characteristic (ROC) curves comparing the performance of naïve (yellow), general (orange) and primed (red) pluripotency gene sets in discriminating "naïve-like" and "primed-like" cell clusters based on the average of their normalised expression levels (left panel). ROC curves comparing the performance of individual naïve pluripotency genes (discriminating "naïve-like" and "primed-like" cell clusters based on normalised expression level), illustrating *Esrrb*, *Klf2* and *Zfp42* as best classifiers of "naïve-like" and "primed-like" clusters (AUC values = 0.928, 0.963 and 0.948 respectively; $p = 10^{-5}$; right panel).

and PU subregions ($p < 2 \times 10^{-27}$). These correspond to the pluripotency factors ESRRB/NR5A2 and KLF2, KLF4 or KLF5 that are highly expressed in naïve pluripotency and downregulated as primed pluripotency is established (Fig. 5a, b). In addition, PU subregions encompassed multiple motifs of pluripotency TFs that remain expressed in EpiSCs, including OCT4 (*Pou5f1*) and SOX2. Collectively, our findings corroborate a possible role for naïve pluripotency factors in regulating DM subregions, and furthermore suggest that PU might be maintained hypomethylated and accessible through hotspot binding of numerous TFs.

In agreement, inspection of the relative enrichment for ESRRB, KLF2, KLF4, KLF5, STAT3 and OSN at PU, DM and INT subregions in ESCs along with other enhancer constituents revealed that all proteins examined were more enriched at PU subregions (Fig. 5c, Supplementary Data 1). While DM subregions displayed lower protein enrichment, we noticed that, of all the TFs evaluated, ESRRB showed the strongest binding at these subregions. Interrogation of independent datasets generated using a modified ChIP-seq protocol with improved site resolution (ChIP-exo)[44] confirmed that DM subregions were indeed highly bound by ESRRB compared to STAT3 and SOX2 (Fig. 5d). These results point to a prominent role for ESRRB in demarcating and possibly regulating DM subregions in ESCs in line with our single-cell analyses (Fig. 4f).

## ESRRB binding inhibits de novo methylation at DM subregions.

To support our conclusion, we asked whether ESRRB is necessary for the enhancer activity of DM-containing SEs in ESCs. For this, we compared the expression fold changes of selected class I (PU containing) and class II (DM-containing only) SE-associated gene candidates in *Esrrb*-depleted ($^{-/-}$) ESCs[45], and *Nanog*$^{-/-}$ ESCs[28] for comparison, relative to control populations. We found that the expression of class II SE-interacting promoters was uniquely sensitive to the depletion of ESRRB compared to class I candidates (Fig. 5e, Supplementary Fig. 7b, c). A similar trend was observed upon depletion of NCOA3 (Supplementary Fig. 7d, e), an essential co-activator of ESRRB in ESCs[46]. *Nanog*$^{-/-}$ ESCs, in contrast, showed no clear segregation between class I and class II candidates (Fig. 5e, right panel). Re-introducing wild-type (WT) *Esrrb* in *Esrrb*$^{-/-}$ ESCs (*Esrrb*$^{WT}$) enhanced the expression of almost all genes tested, with a more pronounced gain at class II compared to class I genes (Fig. 5f, Supplementary Fig. 7f, g). *Esrrb*$^{-/-}$ ESCs were also transfected with a AF-2 mutant (MutAF-2) *Esrrb* form where the ability of ESRRB to recruit co-activators at bound sites is abolished[46]. *Esrrb*$^{MutAF-2}$ cells showed lowered class I and class II gene expression (Fig. 5f, right panel) and could not be maintained in culture. This suggests that overexpressing mutant ESRRB protein might impede the formation of activation protein complexes at SEs, triggering spontaneous differentiation.

Collectively, these findings confirm ESRRB as a potent regulator of self-renewal and transcription in ESCs with class II SE-associated genes being distinctively sensitive to the loss of ESRRB. We note, however, that the expression of these genes further declined in converted EpiSCs (c-EpiSCs) from $-/-$ and control ESCs, implying that the constitutive depletion of ESRRB might destabilise but not fully inactivate DM-containing SEs in ESCs. This agrees with the maintenance of an undifferentiated state in *Esrrb*$^{-/-}$ ESCs, showing no induction of the early *Otx2*, *Fgf5* and *Dnmt3b* differentiation markers and retained expression of *Pou5f1*, *Sox2* and *Nanog* in serum/LIF (see Supplementary Fig. 7h and ref. [45]). To corroborate whether the methylation state of DM subregions was also affected by the loss of ESRRB binding, we focussed on the *Klf4* locus as an example of class II SE-interacting promoters (Supplementary Fig. 7g) and a model gene target of ESRRB[46,47]. Using an assay combining digestion with methylation-sensitive restriction nucleases and locus-specific qPCR amplification[48], we examined CpG sites spanning DM and INT subregions of *Klf4*-associated SE (Fig. 5g). Results revealed an increase in CpG methylation at all DM sites analysed in *Esrrb*$^{-/-}$ compared to control ($^{f/f}$) ESCs. As anticipated, methylation reached similarly high levels at DM and INT sites in c-EpiSCs, where *Klf4* expression is extinguished (Supplementary Fig. 7h). These findings suggest that ESRRB might promote the expression of class II genes, at least partly, by conferring resistance to de novo methylation at DM subregions.

## ESRRB-mediated mediator and POL2 activity at DM subregions.

ESRRB is known to facilitate the recruitment of key TFs and co-activators at ESRRB-bound enhancers in ESCs[46,49–52]. To further elucidate the molecular consequences of ESRRB depletion, we investigated the binding of OCT4, P300 and MED1 at the *Klf4*-associated SE using ChIP-qPCR assays (Fig. 6a; also Supplementary Fig. 8a, b for an extended analysis). No major alteration in the binding profile of either OCT4 or P300 across all regions tested was observed, as previously reported[44]. In contrast, we found that MED1 recruitment was significantly reduced or abolished in the absence of ESRRB. Given the essential role of MED1 in regulating enhancer–promoter interactions in ESCs[8,53], we examined the profile of chromatin interactions at the *Klf4*

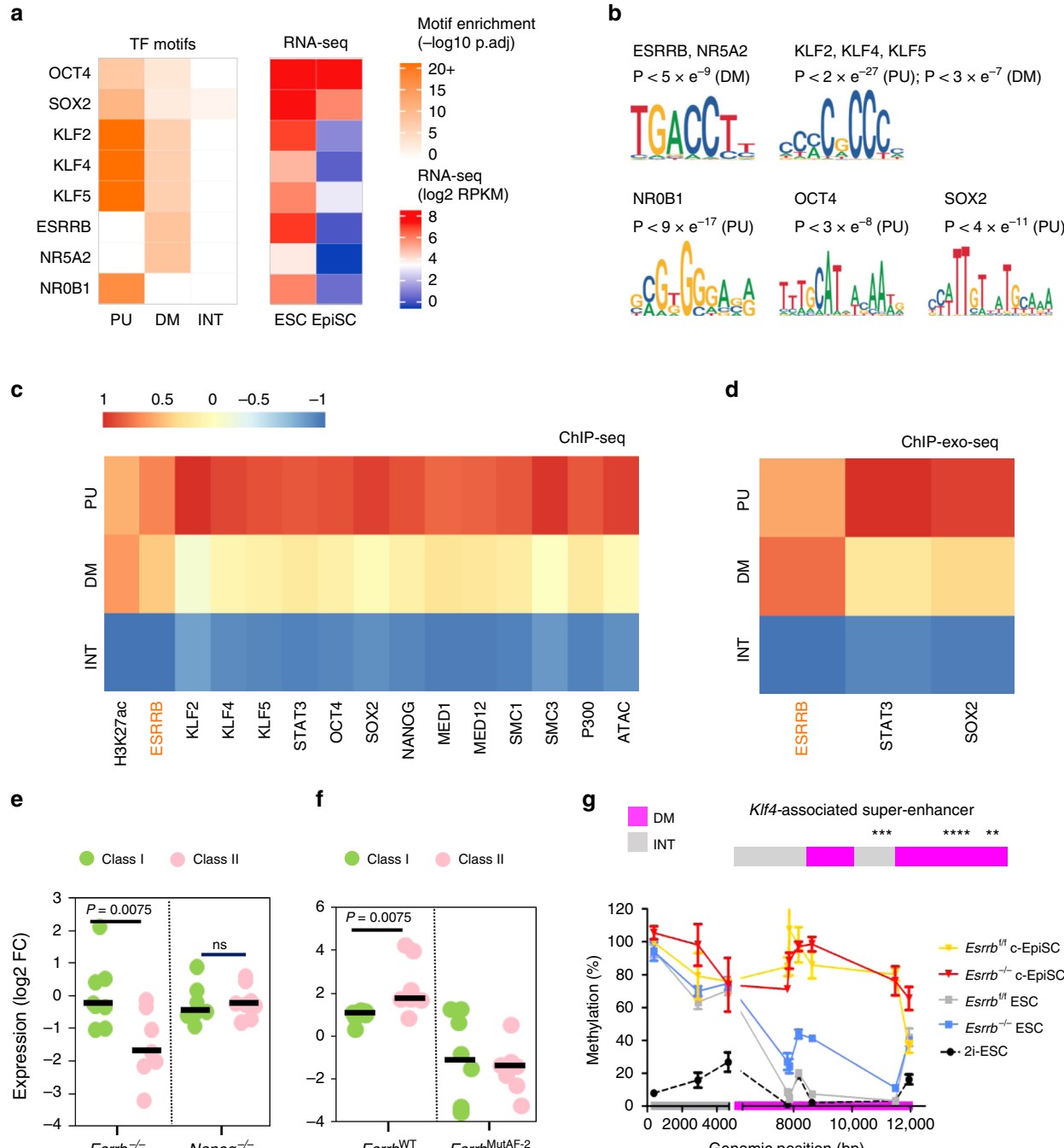

**Fig. 5 DM subregions are most specifically demarcated by high levels of ESRRB binding. a** Transcription factor (TF) motif enrichment in PU, DM and INT subregions, and gene expression of corresponding TFs in ESCs (serum/LIF) and EpiSCs[25]. **b** Motifs and statistical enrichment (hypergeometric test, BH adjusted) for TFs shown in **a**. **c** Heatmap showing the relative enrichment levels (median log$_2$ fold change of ChIP-seq signal (RPKM + 1) over input signal (RPKM + 1)) at SE subregions in ESCs for TFs, co-regulators and chromatin accessibility (ATAC-seq). **d** Heatmap showing relative enrichment levels (calculated as in **c**) at SE subregions in ESCs for ESRRB, SOX2 and STAT3-independent ChIP-exo datasets. Enrichment scores for each feature are scaled by dividing by the mean enrichment score for that feature across all subregions. See Supplementary Data 1 for dataset accession numbers. **e** Expression changes (RT-qPCR) relative to control ESCs of selected class I (*Oct4, Smarcad1, Otx2, Lefty1, Klf13, Med13l, Tet1* and *Nanog*) and class II (*Tfcp2l1, Klf2, Klf4, Klf5, Tdh, Tbx3, Tet2* and *Esrrb*) SE-associated genes in *Esrrb*-depleted ($^{-/-}$) and *Nanog*$^{-/-}$ ESCs. Medians are indicated by bars. Mann–Whitney test was used to compare class I (light green) and class II (pink) expression behaviour. **f** Expression changes (RT-qPCR) in WT or MutAF-2 transfected *Esrrb*$^{-/-}$ ESCs relative to empty vector are shown for selected class I and class II SE-associated genes as in **a**. Data represent three independent experiments (n = 3). **g** CpG methylation at different CpG positions (asterisks) within DM (magenta) and INT (grey) subregions of the *Klf4*-associated SE in *Esrrb*$^{-/-}$ (blue) and control ($^{f/f}$; grey) ESCs and corresponding converted EpiSCs (c-EpiSCs; red and orange, respectively), as well as in control ESCs cultured in 2i/LIF (2i; black dashed). The profile of *Esrrb*$^{-/-}$ ESCs was not analysed in 2i/LIF due to loss of cell viability as previously reported[50]. Data are means ± s.e.m. of three independent experiments (n = 3). Source data are provided as Supplementary Data 10.

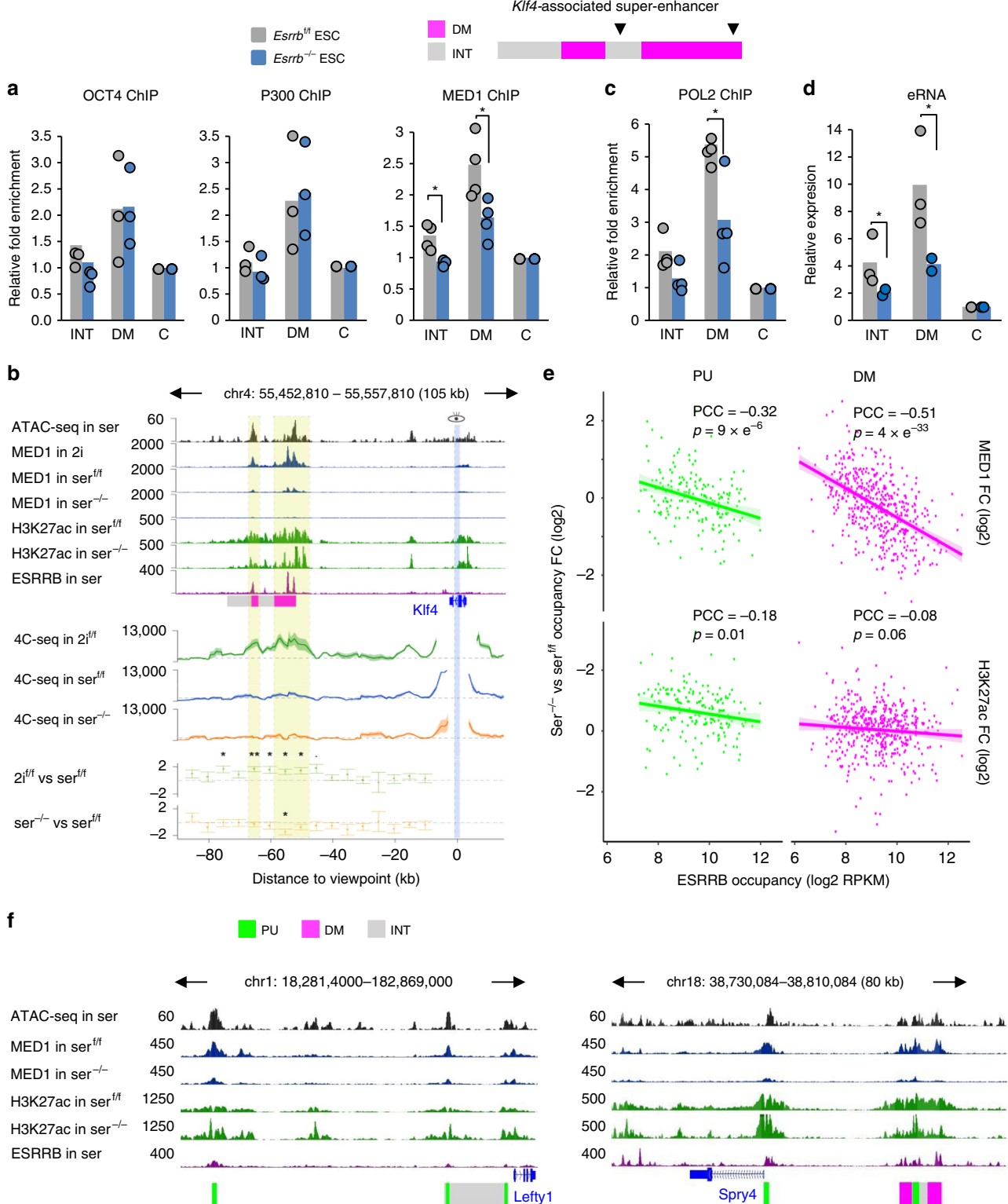

locus using circular chromosome conformation capture (4C-seq) assays in 2i/LIF ESCs (2i) and both control (f/f) and *Esrrb*⁻/⁻ ESCs grown in serum/LIF (ser) with declining levels of MED1 binding (Fig. 6b). Strong interactions were detected in 2i- and weaker interactions in ser-ESCs where cell heterogeneity is most apparent (see Fig. 4). As anticipated, *Klf4* promoter–SE interactions were further lowered in *Esrrb*⁻/⁻ ESCs, particularly at DM subregions (Fig. 6b, Supplementary Fig. 8c). This was

accompanied by reduced POL2 recruitment and expression of eRNA (Fig. 6c, d, Supplementary Fig. 8f, g), further demonstrating decreased enhancer activity at the *Klf4*-associated SE.

To establish the specific dependency of MED1 occupancy on ESRRB binding across all PU and DM subregions, ChIP-seq of MED1 and H3K27ac as a control were performed in ESRRB-depleted and control ESCs (this study). In line with our ChIP-qPCR, strong loss of MED1 occupancy in *Esrrb*⁻/⁻ cells was

**Fig. 6 Destabilised enhancer activity at DM subregions upon depletion of ESRRB in ESCs. a** ChIP-qPCR for OCT4, P300 and MED1 at indicated sites (arrowheads) along the *Klf4*-associated SE in *Esrrb*[−/−] and control ([f/f]) ESCs. Data are expressed as fold enrichment over input and normalised to a flanking control (C) region. Data are represented as means (bar plots) and individual values (dots) of independent experiments (*n* = 3 for OCT4 and P300; *n* = 4 for MED1). *Statistically significant difference (Mann–Whitney test, *p* < 0.05). **b** Top: ATAC-seq in serum/LIF (ser), MED1 in 2i/LIF (2i) and ESRRB in serum/LIF (ser) (Supplementary Data 1) along with MED1 and H3K27ac in control (ser[f/f]) and *Esrrb*[−/−] ESCs grown in serum/LIF (ser[−/−]) at the *Klf4* locus. Y-axis indicates RPKM and starts at 0; bottom: 4C-seq interactions between the *Klf4* promoter viewpoint (blue band) and SE in control ESCs grown in 2i/LIF (2i[f/f]) or in serum/LIF (ser[f/f]), and in serum/LIF *Esrrb*[−/−] ESCs (ser[−/−]). Y-axis indicates RPKM and starts at 0. The bottom two tracks show the 4C-seq interaction fold changes (log2) in 2i[f/f] and ser[−/−] compared to ser[f/f] in bins of 5 kb. Promoter–SE interactions at the ESRRB/MED1 occupied DM regions (yellow bands) are significantly stronger in 2i and weaker in ser[−/−]. *p* < 0.05, **p* < 0.01; DEseq2; (*n* = 3 independent experiments). **c** ChIP-qPCR for POL2 at the same sites as in **a** in *Esrrb*[−/−] and *Esrrb*[f/f] ESCs. Data are expressed as fold enrichment over input and normalised to control (C) region. Data are represented as means and individual values of independent experiments (*n* = 4). Statistically significant difference (Mann–Whitney U test, *p* < 0.05). **d** Expression of eRNA at the same sites as in **a** in *Esrrb*[−/−] and *Esrrb*[f/f] ESCs. Data are normalised to control (C) region and are represented as means and individual values of biological replicates; (*n* = 2 for *Esrrb*[−/−]; *n* = 3 for *Esrrb*[f/f]). *Statistically significant difference (Mann–Whitney test, *p* < 0.05). **e** MED1 and H3K27ac occupancy changes in *Esrrb*[−/−] (ser[−/−]) compared to control (ser[f/f]) ESCs. PCC, Pearson's correlation coefficient; (*n* = 2 independent experiments). **f** ChIP-seq tracks of MED1 and H3K27ac in *Esrrb*[−/−] (ser[−/−]) and control (ser[f/f]) ESCs, as well as ESRRB and ATAC at *Lefty1* and *Spry4* -associated SEs. Source data are provided as Supplementary Data 10.

confirmed genome wide (Fig. 6e). Importantly, the degree of MED1 loss correlated significantly with ESRRB occupancy levels in WT ESCs, particularly at DM subregions (PCC = −0.51, *p* = $4 \times 10^{-33}$). In contrast, ESRRB depletion did not significantly affect the H3K27ac levels of PU and DM subregions. As MED1 can be recruited by multiple TFs[54,55], the relationship between ESRRB occupancy and MED1 loss was also examined in the context of OCT4 binding as control. Using a linear regression model (see Methods section; Supplementary Data 9), we found no significant association between OCT4 binding scores in ESCs and the loss of MED1 upon ESRRB depletion (Student's *t*-test, *p* = 0.27), highlighting the specificity of ESRRB–MED1 association in our model system.

Collectively, our results dissect SEs into ESRRB-dependent (DM) subregions that most specifically regulate ESC-specific pluripotency genes and become decommissioned in the primed state. These SE subregions are highly bound by ESRRB in naive ESCs and display strong loss of MED1 occupancy and enhancer activity in ESRRB-depleted cells concomitant with the acquisition of CpG methylation (e.g., *Klf4*). In contrast, PU subregions retain MED1 occupancy in ESRRB-depleted cells. These regions are associated with genes that maintain or even gain expression in primed pluripotency (e.g., *Lefty1*, Fig. 6f, left panel). Lastly, partial decommissioning of SEs that contain both PU and DM subregions explains the lowered but not completely lost expression of some pluripotency genes during early embryonic development (e.g., *Spry4*, Fig. 6f, right panel).

## Discussion

SEs are defined as clusters of enhancer units located within large domains of H3K27ac deposition and cell-type-specific TF binding. How these domains are organised and to which extent enhancer units within SEs are functionally equivalent is still the matter of debate[44,56–60]. In our study, we delved into these questions in the context of ESCs, particularly at the earliest steps of differentiation where the exit from naive pluripotency is regulated. Under serum/LIF conditions, we show that enhancer units within SEs are linked together by methylated interstices (INT), as previously suggested[61]. The focal unmethylated subregions of SEs coincide with the binding of TFs and co-regulators (e.g., MED1 and POL2) where SE–promoter interactions preferentially assemble, in agreement with the concept of hub enhancers[62]. Moreover, we find that the prevalence of chromatin interactions and eRNA transcription at unmethylated over methylated regions is conserved under 2i/LIF conditions, which enforce a globally hypomethylated state in ESCs (Supplementary

Figs. 3h–j and 8d). This suggests that the organisation of SEs is largely imposed by cell-type-specific TFs, most likely counteracting CpG methylation at binding sites under permissive conditions[14,63–66]. Unexpectedly, however, we uncover pronounced differences in the dynamics of CpG methylation and chromatin configurations amongst SE enhancer units as unveiled at the onset of ESC differentiation. Functionally, we show that enhancer units within SEs partition into two subtypes (i.e., PU and DM) that follow independent fates during the naive-to-primed pluripotency transition. While PU subregions remain hypomethylated, highly accessible and hotspots of protein binding in ESCs and EpiSCs, DM subregions are targeted by de novo methylation and loose their enhancer signatures (e.g., OCT4 binding, H3K27ac and ATAC-seq signals) in the primed cells. Hence, while PU and DM enhancer units are both engaged in ESCs, they become constitutively active (PU) or decommissioned (DM) in EpiSCs, as further established in the peri-implantation epiblast in vivo.

Remarkably, we find that PU subregions are not detected across all ESC SEs and most specifically regulate the expression of a core set of genes shared by naive and primed cells. This evokes a pivotal role for PU subregions in the upholding of pluripotency during this key developmental transition. Of interest, hotspots of TF binding were also reported within lineage-specific SEs with a prevalent role at the onset of progenitor differentiation[67,68]. Thus, PU-like subregions might similarly operate in other cell state transitions at different stages of development. Whether PU subregions mapped in pluripotent cells are subsequently inactivated upon gastrulation, as suggested by their methylated profiles in somatic cells (Supplementary Fig. 4g), and what are the molecular pathways protecting their activity prior to lineage specification are still to be fully delineated. Of relevance, previous studies suggest that selective naive enhancers transiently escape decommissioning via the binding of distinct TFs whose expression is regionalised during the patterning of the epiblast[69,70]. Conversely, we observe that a large number of germ-layer-associated TF motifs (e.g., HOX, IRX, NKX, OLIG, PAX and SOX) are enriched within INT (methylated) regions of SEs, pointing to the presence of "latent" lineage-specific enhancer units (or seed enhancers[10]). While these putative enhancer units are most likely inactive in pluripotent cells, they might become unveiled in a tissue-specific manner upon CpG demethylation in due course of development.

In contrast to PUs, we show that DM subregions are demarcated by a high-level of ESRRB binding. Concurringly, ESRRB's cognate binding motif is strongly enriched at DM relative to PU subregions in contrast to other pluripotency TF motifs. Functionally, we establish that DM enhancer units regulate the expression of pre-

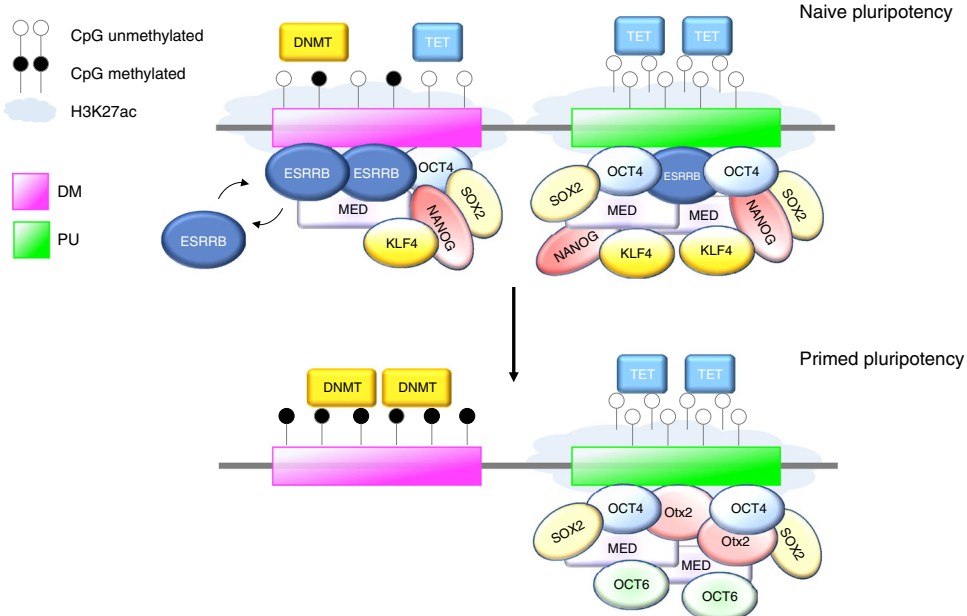

**Fig. 7 Proposed model depicting the role of ESRRB at SEs in pluripotent stem cells.** In naive pluripotent cells, the dynamic expression and binding of the nuclear receptor ESRRB instigates a metastable state at DM subregions (magenta) by counteracting DNMT3s activities as reflected in cell-to-cell CpG methylation heterogeneity. In contrast, PU subregions (green) consist of hotspots of core and naive pluripotency TF binding and remain stably unmethylated. Upon ESRRB depletion and/or during the establishment of primed pluripotency, DM subregions are rapidly and selectively destabilised with the loss of mediator occupancy and consolidation of CpG methylated state, leading to their decommissioning in primed epiblast cells. In contrast, PU subregions remain unmethylated and highly bound by core and primed pluripotency TFs, indicative of their continued enhancer activity. The partitioning of enhancer units within SEs as constitutively active (PU) or decommissioned (DM) proposes a mechanism by which the pluripotency transcriptional programme can be partially reset during the naive-to-primed transition, preserving pluripotency as cells prepare for subsequent differentiation. Core (OCT4 and SOX2), naive (ESRRB, KLF4 and NANOG) and primed (OCT6 and OTX2) pluripotency TFs are represented along with the mediator protein complexes (MED), DNMT and TET enzymes.

implantation gene modules, and are uniquely sensitive to the loss of ESRRB that underlies or triggers the exit from naive pluripotency[48,50]. Focussing on the *Klf4*-associated (DM-containing) SE as a model locus, we demonstrate that ESRRB depletion in ESCs is sufficient to impede the loading of the mediator complexes at *Klf4*-associated SE, reduces the expression of its target gene, and promotes CpG methylation at DM subregions. This might involve methylation spreading from DNMT3B highly bound INT regions owing to the processive activity of DNMT3B enzymes[66,71]. ESRRB-dependent MED1 recruitment is further confirmed across all SEs genome wide, particularly at DM subregions, and is essential for maximal transcriptional activation by promoting class II SE–promoter interactions. Accordingly, we find that chromatin interactions at the *Klf4* locus are destabilised upon ESRRB depletion, concomitant with a reduction in POL2 recruitment and eRNA production. These findings corroborate knowledge of nuclear receptor-mediated gene activation mechanisms[72–74], and furthermore are supported by the ability of ESRRB to interact with mediator and POL2 complexes in ESCs[46,49,51]. Given the importance of ESRRB in stabilising the recruitment of these complexes, possibly via its co-activator NCOA3, it will be of interest to study the role of ESRRB-NCOA3 in the formation of phase-separated condensates recently identified as key activation domains, particularly at SEs[54,55].

Another most interesting feature of DM subregions is their varying levels of CpG methylation as revealed in individual ESCs. While both methylase and demethylase enzymes are co-recruited to DM subregions, we show that DNMT3s rather than TET activities drive methylation variance at SEs. Besides DNMT3s, we reveal that cell-to-cell DM epigenetic heterogeneity closely associates with the variable expression of *Esrrb*, which in turn is

under the control of a DM-containing (class II) SE and dynamically methylated in cells initiating differentiation (see Fig. 4c and ref. [48]). Given ESRRB's ability to access ERRE binding sites within methylated regions[75] and inhibit de novo CpG methylation upon binding (this study), we propose that a balance between DNMT3s and ESRRB activities instigates a metastable state at DM subregions prone for decommissioning upon exit from naive pluripotency (Fig. 7). This metastable state is thought to be resolved upon *Esrrb* silencing and subsequent consolidation of CpG methylation at these sites, facilitating the dismantling of the pre-implantation transcriptional programme as pluripotency is safeguarded post-implantation. In line with this model, depletion of DNMT3s is known to delay *Esrrb* extinction and the exit from naive ESC pluripotency (ref. [76] and our unpublished observations). It is worth noting that the action of ESRRB is not restricted to SEs but most likely extends to a subpopulation of TEs that are targeted by hypermethylation in EpiSCs and similarly sensitive to the loss of ESRRB in ESCs (Supplementary Fig. 9). Thus, our study highlights the pivotal role of ESRRB in regulating and partitioning naive enhancers during pluripotency state transitions[50,75], and furthermore offers mechanistic insights into the nature of the molecular events that follow the loss of ESRRB during early development.

## Methods

**Datasets**. Gene Expression Omnibus accession numbers for the sequencing data generated in this paper are GSE124476 (BS-seq in EpiSCs), superseries GSE139189 (MED1, H3K27ac ChIP-seq and 4C-seq in *Esrrb*⁻/⁻ and control ESCs). All other publicly available datasets used are specified in Supplementary Data 1. The mm9 reference mouse genome was used for our study.

**Cell culture.** Mouse ESCs were routinely cultured on 0.1% gelatin coated plates and maintained in Glascow Minimum Essential Medium (GMEM) media supplemented with 10% fetal bovine serum (serum), MEM non-essential amino acids, beta-mercaptoethanol, L-glutamine, sodium pyruvate, sodium bicarbonate, penicillin/streptomycin, LIF (prepared in-house) and the appropriate drug selection (serum/LIF conditions). Where mentioned, ESCs were adapted to serum-free culture conditions using either N2B27 or chemically defined medium (CDM[77]) supplemented with 1 μM PD0325901, 3 μM CHIR99021 and LIF (2i/LIF conditions). c-EpiSCs and embryo-derived EpiSCs were cultured in N2B27 (ref. [26]) or CDM[25], respectively, both supplemented with 20 ng/mL activin A and 12 ng/mL fibroblast growth factor 2 (FGF2). Mouse $Esrrb^{-/-}$ ESCs[45], $Nanog^{-/-}$ ESCs[28] and matching control populations have been previously described. $Ncoa3^{-/-}$ and control ESCs were derived from mutant and WT B6/129 mice and kindly provided by Austin Cooney. For the generation of rescued $Esrrb^{-/-}$ ESCs, cDNA encoding $Esrrb$ WT or AF-2 point-mutant[46] form was cloned into the pPyCAGIP vector, and one million of cells transfected with Lipofectamine 2000 and 2 μg of either of these two vectors or an empty vector (control). Twenty-four hour post-transfection 1 μg/mL puromycin was added for selection and after 8–10 days of culture individual ESC clones were isolated and expanded indefinitely under selection.

**Conversion of ESCs into EpiSCs.** R1-ESCs (ATCC) used for conversion were cultured in 2i/LIF or serum/LIF. To induce conversion into the primed EpiSC state, ESCs were trypsinized and replated into CDM supplemented with FGF2 (12 ng/ml) and activin A (20 ng/ml), on serum-coated cultures plates[27]. Passage was performed after 4–5 days using collagenase II treatment. Cells were considered as stably converted (c-EpiSCs) after at least three passages in the presence of FGF2 and activin A.

**RT-qPCR.** Total RNA was isolated and DNaseI-treated using the RNeasy mini kit (Qiagen). For eRNA detection, total RNA was isolated using Trizol (Invitrogen) and DNAse-treated was performed on purified RNA using TURBO DNA-*fre* Kit (Invitrogen). Samples were reverse-transcribed using SuperScript III (Invitrogen) and random primers following the manufacturer's instructions. For quantification, cDNA (or DNA) samples were amplified with SYBR Green PCR Mastermix (Sigma or Applied Biosystems), using a StepOne™ System (Applied Biosystems). Data were normalised using the geometric mean of $Sdha$ and $Pbgd$ for conversion experiments or $S17$, $L19$ and $Gapdh$ for mutant, and matching control ESCs. Primers used in RT-qPCR assays are listed in Supplementary Data 6.

**Chromatin immunoprecipitation.** ChIP was performed as previously described[78] with minor modifications outlined below. Chromatin was fixed with 1% formaldehyde (Sigma-Aldrich) for 10 min and sonicated on a bioruptor (Diagenode) to produce fragments of 100–500 bp, and ChIPs performed with Protein G-coupled magnetic Dynabeads (Invitrogen) and the following antibodies: 8 μg MED1 (A300-793A Bethyl Laboratories), 5 μg OCT4 (sc-8628 SantaCruz), 5 μg p300 (sc-585 SantaCruz), 5 μg POL2 (Clone 8WG16 MMS-126R, Covance) and 5 μg H3K27ac (Ab4729 Abcam). The amounts of chromatin (protein) used in each ChIP were as follows: 400 μg (OCT4), 500 μg (P300), 500 μg (H3K27ac) and 800 μg (MED1 and POL2). Following washes of bound DNA–protein complexes, DNA was eluted in 1% sodium dodecyl sulfate (SDS) and treated with 40 ng/μl RNaseA following 0.2 μg/μl Proteinase K. After phenol/chloroform purification, DNA was then precipitated at −20 °C with 20–30 μg GlycoBlue carrier (Invitrogen), 1/10 volume of 3 M NaAc and 2 volumes of 100% ethanol. Resuspended pellets were used for qPCR or for generation of libraries for sequencing (MED1 and H3K27ac). Sequencing libraries were prepared using the NEBNext® Ultra™ DNA Library Prep Kit and Multiplex Oligos (New England Biolabs) from 5 ng of DNA. Following analysis on an Agilent Bioanalyzer libraries were pooled and sequenced on an Illumina Genome Analyzer II (Illumina). Quality of the sequenced reads was assessed using the FASTQC program (Babraham Bioinformatics). Primers used in ChIP-qPCR assays are listed in Supplementary Data 6.

**5mC Analysis by restriction enzyme digestion.** Genomic DNA was extracted using the DNeasy kit (Qiagen). A total of 1 μg of eluted DNA was diluted in 17 μl of 20% TE buffer. For each set of enzyme reaction, 2 μl of appropriate restriction enzyme buffer was added to the diluted DNA. A volume of 9.5 μl of the mixture was then transferred to a separate tube, serving as undigested control. A volume of 0.5 μl (5U) of enzymes was added to the remaining DNA mixture. Both digestion reactions and undigested control were incubated overnight in 37 °C incubator. A volume of 95 μl of 20% TE buffer was then added to both digested and undigested samples, which were then proceeded with qPCR analysis. Enzymes (methylation sensitive) used in this study are BsaAI, SsiI, HpaII and Hin6I, and qPCR primer sequences are listed in Supplementary Data 6.

**Western blots.** Cells were lysed for 30 min on ice into RIPA buffer (150 mM NaCl, 1% NP-40, 0.5% NaDeoxycholate, 0.1% SDS, 50 mM Tris-HCl pH8.0) in the presence of protease and phosphatase inhibitors (Pierce). Proteins were quantified using BCA assay (Pierce). A total of 10–15 μg of proteins were charged on pre-cast polyacrylamide gel 4–15% (Biorad) for 1 h run at 100 V. Transfer was then performed on Trans-Blot Turbo (Biorad) for 7 min on a PVDF membrane

(Hybond-P, GE Healthcare). After blocking in TBS-Tween20 0.01% (TBS-T) with either 4% non-fatty milk or 5% BSA, membranes were incubated overnight at 4 °C with primary antibodies. After washes in TBS-T, membranes were incubated with secondary antibodies for 1 h, washed and revealed with ECL2 western blotting substrate (Pierce). Chemiluminescent signals were captured using Chemidoc Touch imaging system (Biorad) and then analysed with ImageJ (imagej.nih.gov/ij). Signals were normalised to H3 or Actin. Western blots were repeated at least three times. Antibodies used were: ESRRB (R&D H6705; 1:1000), NCOA3 (SantaCruz Sc9119; 1:1000), NANOG (Abcam, 80892; 1:1000), ACTIN (Sigma, A5441; 1:5000) and H3 (Abcam ab1791; 1:10,000). Uncropped and unprocessed scans of blots are included in Supplementary Data 10.

**4C-sequencing.** $Esrrb^{f/f}$ and $Esrrb^{-/-}$ ESCs were cultured in serum/LIF or 2i/LIF (control cells only), and three biological replicates were employed per condition. 4C-seq experiments were performed as previously[50] with minor modifications. Briefly, 10 million cells were crosslinked for 13 min with 2% paraformaldehyde in ESC culture medium, quenched with glycine and lysed in 15 ml lysis buffer (10 mM Tris pH 7.5, 10 mM NaCl, 0.2% NP-40, 1× protease inhibitors) for 30 min at 4 °C. Nuclei were then incubated with 0.25% SDS for 30 min in NEB buffer3 followed by Triton X-100 treatment for 30 min, both at 37 °C. Nuclei were then digested with 700 U DpnII enzyme (NEB) overnight at 37 °C. The enzyme was inactivated at 65 °C for 15 min followed by "in nuclei" ligation at 16 °C with 2000 U T4 ligase (NEB). Samples were then treated with protease K and RNAseA, and DNA was purified by phenol–chloroform (Sigma). DNA was further digested with 50 U BfaI, and purified with QIAquick PCR purification columns followed by a second ligation at 16 °C. Next, 1000 ng of 4C-seq library was amplified with bait-specific inverse primers[50] and using the Expand Long template PCR system (Roche 11759060001) for 28 PCR cycles. PCR products were then purified and 50 ng DNA was used for library preparation using KAPA Hyperprep kit (Roche) and five PCR cycles. Libraries were sequenced on the Illumina NextSeq 500 (Illumina) to obtain paired-end sequences of 50 bp.

**In silico analysis of CpG methylation data.** For each WGBS sample, CpG information have been filtered to keep only those for which we had a sufficient coverage of at least 7 reads (apart from GSM1904118 and GSM1904112 datasets). HMM analysis has been applied on each chromosome independently. Genome was split in segments containing CpGs spaced by <1 kb. Segments containing <10 CpGs have been filtered out. Parameters of HMM analysis on the remaining CpGs were initialised using viterbiEM function of tileHMM R-package; CpGs were considered to be in two different states: methylated or unmethylated. Corresponding HMM model has been applied for CpGs states call on the whole chromosome. Then, in ESC (serum/LIF) and EpiSC independently, segments of SEs containing at least four consecutive unmethylated CpGs were collected (with a coverage of at least 7 reads and distant of <1 kb). Intersection of EpiSC and ESC unmethylated segments were defined as PU regions. Segments containing at least four consecutive CpGs that were unmethylated in ESCs and methylated in EpiSCs were defined as DM regions. Segments of SE regions located between PU and DM segments were defined as INT segments. Coordinates and CpG methylation information on the different samples (in vitro and in vivo) can be found in Supplementary Data 3. In Figs. 1a and 2a, CpG methylation is presented on a −1 to 1 scale, in order to visualise their unmethylated (negative values) or methylated (positive values) state as determined by HMM. The height of the bar indicates the percentage of methylation within reads covering each CpG, with positive values representing the extent of methylation, and negative values, the extent of demethylation (−1 +% methylation).

**ChIP-seq data processing and analysis at SE subregions.** All ChIP-seq datasets were processed from raw reads (Fastq files) to filtered, mapped and deduplicated reads (bam files) through a standardised pipeline. This pipeline involves: adaptor removal, low-quality read trimming and filtering using Trimmomatic; alignment to mm9 reference genome with Bowtie2; duplicate read marking and removal with Picard Tools. ChIP-seq coverage plots were produced as follows (assuming the total read count has been calculated at base-pair resolution for a set of mapped reads, scaled by sequencing library depth and normalised to scaled coverage from input DNA library): for each of a set of genomic regions of interest, the region is split into a fixed number (1000) of windows of equal width, and the average coverage across each window is computed; the average of each window's coverage is then computed across all regions of interest.

**ATAC-seq data processing and accessibility analysis.** Chromatin accessibility measured by ATAC-seq in ESCs grown in 2i/LIF (2i), serum/LIF (ser), and in EpiSCs was downloaded from published data (Supplementary Data 1) and mapped with Bowtie2. Low-quality mapping reads (MAPQ < 10) and duplicated reads were omitted for further analysis. ATAC-seq peaks were called with MACS2 version 2.1.1.20160309, with a $q$-value threshold of 0.01 and using whole cell extract (WCE) (input) as control. The number of ATAC-seq peaks intersecting SE subregions in 2i-ESCs, ser-ESCs or EpiSCs was computed using the summarizeOverlaps function from the GenomicRanges R-package. When multiple SE subregions overlapped the same ATAC-seq region, the SE subregion with highest

overlap was assigned. The average chromatin accessibility level per SE subregion was computed with the featureCounts function in the Rsubread package and normalised to RPKM after subtracting the read counts from the WCE (input) data. Data are shown in Supplementary Data 3.

**RNA-seq analysis**. Gene expression values (average RPKM of at least two replicates) were taken from publicly available RNA-seq datasets (Supplementary Data 1). Genes with an expression value of 1 RPKM or more in ESCs grown in serum/LIF ($n = 11,087$) were considered expressed.

**Promoters of closest expressed genes**. Promoters were defined as the 5 kb window surrounding an annotated transcription start site. Gencode GRCm37 (version M1) gene annotation was used. The distance from a SE to the promoter of the closest expressed gene was determined with the "*distanceToNearest*" function in the R-package *GenomicRanges* Only promoters of known genes were considered.

**4C-seq data analysis**. 4C forward or reverse PCR primers from paired-end sequenced FASTQ files were trimmed with cutadapt allowing a 10% mismatch: cutadapt -g [primer_seq] -O [primer_length −2] -e 0.1 --discard-untrimmed. Trimmed reads were mapped to the mm9 reference genome using Bowtie2 with the option "very-sensitive" in single-end mode. Low-quality reads (MAPQ < 10) were discarded. The FourCSeq package was used to map reads from the forward and reverse PCR primer to valid restriction sites. Reads were normalised compared with DESeq2 discarding reads that mapped in trans. For visualisation, counts were smoothed using a running mean with $k = 7$ bins. Differential analysis was done by DEseq2 using the local dispersion fit, after summing read counts in bins of 5 kb surrounding the viewpoint (up to 1 Mb up- and downstream). Bins with coverage <1000 RPKM were discarded for differential analysis. Data are shown in Supplementary Data 7.

**Capture Hi–C analysis**. High-resolution capture Hi–C (CHiC) studies (Supplementary Data 2) were used to map significant promoter–SE interactions in ESCs grown under serum/LIF (ser) or 2i/LIF (2i). Data from Joshi et al. were previously mapped with BWA MEM to the GRCm37 (mm9) reference genome. The other datasets were mapped with HiCUP. For all datasets, PCR duplicates, read pairs mapping to the same restriction fragment (self-ligation) and pairs with low mapping quality (MAPQ < 10) were removed. For the DpnII data (Joshi)[15] and NcoI data (Sahlen)[16], four consecutive restriction fragments were merged into a pseudo-fragment to increase the read count and confidence per called interaction. We used the CHICAGO pipeline for CHiC[17] to call significant interactions between these pseudo-fragments in the ser-ESC or 2i-ESC state with a default threshold (score >= 5). Additionally, interactions with coverage of <5 reads (geometric mean of the replicates) were discarded. CHICAGO takes the geometric mean of the pairwise interactions counts when multiple replicates are available. The much lower library depth of the second replicate (Supplementary Data 2) causes lower read counts on average and a much smaller number of significant interactions. Given that the CHiC libraries from Joshi et al. can be treated as independent replicates with the same effective resolution, we decided to merge the interaction read counts for the Sahlen replicates prior to running the CHICAGO pipeline.

**CHiC library normalisation and correlation analysis**. Interaction frequencies were normalised using DESeq2. Because CHiC is based on proximity ligation, loci in close proximity of the capture bait have higher read counts and more variance compared to more distal loci. To mitigate this effect, we applied the DESEq2 normalisation in four distance categories (<25 kb, 25–100 kb, 100–300 kb and >300 kb) following an approach we used earlier[50]. Next, we computed the pairwise Spearman correlation coefficient between the promoter–SE/promoter–SE subregions interactions in library X and library Y.

**Promoter–subregion interaction frequency at SE subregions**. CHiC interaction frequency between promoters and SE subregions was computed at the native 1 restriction fragment resolution (DpnII or NcoI) for all promoter–SE pairs that had a significant interaction. SE subregions smaller than 500 bp were discarded since they often have too few overlapping restriction fragments to enable a robust analysis. Statistical differences per subregion class (PU, DM and INT) were assessed by a linear regression model that accounts for the two major confounders: the number of capture baits per subregion and the promoter–subregion distance (log2).

**Predicting expression changes using BS-seq and capture Hi-C**. We hypothesised that changes in CpG methylation would mostly affect the gene expression of the strongest interacting promoters (Fig. 3a). In other words, expected gene expression changes are a function of the CpG methylation change from ESCs (serum/LIF) to EpiSCs, as well as the CHiC interaction strength (and its changes). Therefore, we estimated the expected expression change $\Delta X = \log2(\text{normalised CHi–C reads}) \times (\%CpG \text{ ESC} - \%CpG \text{ EpiSC})$. The analysis was restricted to PU and DM SE subregions.

**TF motif analysis**. We used Gimme motifs to find TF motifs that are statistically enriched in the PU, DM or INT SE subregions. Since the SE subregions are typically quite broad, we partitioned each SE subregion into equally spaced regions with a length of 291 bp; the median length of the ATAC-seq peaks. To determine a threshold for TF motif presence/absence, 50,000 regions of 291 bp were randomly sampled from the genome. For each motif, the 99% was used as a cut-off, leading to an empirical false discovery rate of 0.01 (gimme threshold). Next, we counted the number of present motifs in the PU, DM and INT subregions of SEs relative to the union of the regions and applied a hypergeometric test. p-Values were adjusted for multiple testing using Benjamini–Hochberg correction.

**sc-BS-seq and RNA-seq processing**. Processed scM&Tseq data[30] were obtained from the Gene Expression Omnibus (accession GSE74534). Segments of SE regions were mapped to mm9 coordinates using UCSC liftOver tool (https://genome.ucsc.edu/cgi-bin/hgLiftOver). For each SE subregion type (PU, DM and INT), mCpG/total-CpG ratio was computed from all reads mapping to any CpG site within an SE segment of the corresponding type. Complete linkage hierarchical clustering was performed on all available ESCs, using the squared differences between the total DM mCpG/total-CpG averages of each cell. This clustering was used to define two clusters of ESCs: one cluster included all 2i-ESCs and a subset of the serum-ESCs (which we defined as "naive-like" ESCs); the other cluster contained only serum-ESCs (which we defined as "primed-like" ESCs). Differential gene expression analysis was performed using Limma to compute empirical Bayes moderated t-statistics from linear models fitted to RNA-seq read counts for serum-ESCs by the DM methylation cluster to which the corresponding cell had been assigned. DNA methylation profiles (Fig. 4c, Supplementary Fig. 6b) were created using Loess smoothing of estimated methylation at each CpG locus. For any given ESC treatment condition, the variance in CpG methylation level for each individual cell analysed of that condition was computed among all different SE subregions. The distributions of these methylation variances are shown as box plots (Fig. 4e, Supplementary Fig. 6a).

**Evaluation of ESC classification (ROC curves)**. sc-RNA-seq read counts for each mapped gene were normalised by median centring and scaling to a standard deviation of 1. ROC curves were prepared by plotting sensitivity against 1-specificity. In this context, sensitivity is the proportion of all "naive-like" cells among the top-ranking $n$ according to the signature of interest; 1-specificity is the proportion of all non "naive-like" cells among the top-ranking $n$ according to the signature of interest. A signature score is either the normalised read count in the given cell for a single gene, or the mean of normalised read counts of a set of genes in the given cell. AUC was computed through numeric integration of the corresponding ROC curve.

**Predicting DM/PU status based on epigenetic features**. To evaluate predictive power of epigenetic features to classify SE subregions as PU or DM, logistic regression models were fitted with region class as a binary outcome (DM vs PU) and each of a set of features as quantitative predictor variables: average ChIP-seq enrichment for TET1, DNMT3A/B, H3K4me1, H3K4me3, H3K27ac, average ATAC-seq signals in serum/LIF ESCs and CpG density. Models were fitted using the generalised linear model function implementation 'glm' in R. Model coefficient estimates, standard errors, t-statistics and corresponding p-values are provided in Supplementary Data 8. Positive coefficient estimates imply subregions with increased values for the corresponding feature have increased probability of being classed as DM (as opposed to PU).

**Impact of OCT4 binding on ESRRB–MED1 relationship**. MACS2 was applied to call peaks from serum/LIF ESC OCT4 ChIP-seq study analysed for Fig. 3e. SE subregions were assigned an OCT4 binding score using average log(ChIP/control) enrichment for any peaks overlapping the SE subregion, or assigned a score of 0 if no peaks overlapped. A linear regression model was fitted to $\log_2$ fold change of MED1 ChIP-seq enrichment in $Esrrb^{-/-}$ relative to $Esrrb^{fl/fl}$ ESCs as a quantitative outcome variable, with serum/LIF ESC ESRRB ChIP enrichment, subregion class (DM vs PU) and OCT4 binding score as predictor variables. Model coefficients, t-statistics and p-values were obtained from the fitted linear model using the 'summary.lm' function in R, and are provided in Supplementary Data 9. Negative coefficient estimates imply SE subregions with higher values for the corresponding feature show a greater decrease in MED1 DNA-binding signal following ESRRB depletion.

**Visualisation of genomic data**. ATAC-seq and ChIP-seq bigwig tracks were prepared using deeptools "bamCoverage", with parameters "binsize" = 10, "normalizedUsing" = RPKM and "extendReads" = 200 (or fragment size in the case of paired-end sequencing). Tracks where visualised on the Washu epigenome browser.

**Data representation**. *Box plots*: Centre lines show the medians; box limits indicate the 25th and 75th percentiles as determined by R software; whiskers extend

1.5 times the interquartile range from the 25th and 75th percentiles, outliers are represented by dots.

*Violin plots*: White dots show the medians; box limits indicate the 25th and 75th percentiles as determined by R software; whiskers extend 1.5 times the interquartile range from the 25th and 75th percentiles; polygons represent density estimates of data and extend to extreme values.

**Reporting summary**. Further information on research design is available in the Nature Research Reporting Summary linked to this article.

## Data availability

New datasets generated in this study has been deposited in GEO: WGBS in EpiSCs (GSE124476), H3K27ac, MED1 ChIP-seq and 4C-seq in $Esrrb^{-/-}$ and control ($^{f/f}$) ESCs (superseries GSE139189). The source data underlying Fig. 3d and Supplementary Fig. 5a, c, e, f; Supplementary Fig. 7e, g, h; Fig. 5g; Fig. 6a and Supplementary Fig. 8b; Fig. 6c and Supplementary Fig. 8f; Fig. 6d and Supplementary Fig. 8g; Supplementary Fig. 7b, d, c, f; Supplementary Fig. 8c are provided as Supplementary Data 10.

## Code availability

Codes are available at https://github.com/edcurry/esc-se-regions.

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

## Acknowledgements

We are grateful to Austin Smith, Hitoshi Niwa, Ian Chambers and Austin Conney for providing ESC lines constitutively KO for *Esrrb*, *Nanog* or *Ncoa3*. Thanks to Tony Bou-Kheir, Megha Prakash-Bangalore, Onkar Joshi, James Flanagan and John Galon for their technical and/or bioinformatics assistance. Thanks to the Sequencing Facility at the Radboud Institute for Molecular Life Sciences. Thanks also to Michelle Percharde, Helle Jorgensen, Wei Cui and Tristan Rodriguez for discussions and/or critical reading of the manuscript, and to all members of the Epigenetics and Development group. This work was supported by the Medical Research Council—U.K. (MR/K500793/1 and MR/K00090X/1; E.B.), the Imperial NIHR Biomedical Research Centre—U.K. (E.W.C.), ERC grant ERC-2013-AdG no. 339431—SysStemCell (W.M., Y.A. and H.G.S.), ANR Programme Investissements d'Avenir REVIVE ANR-10-LABX73 (L.J., V.B. and A.J.); the Fundação para a Ciência e a Tecnologia—Portugal (SFRH/BD/7024/2010; R.A.T.), Genesis Research Trust—U.K. (K.H.T.M.), Imperial College President PhD scholarship—U.K. (R.A.d.S.), Netherlands Organisation for Scientific Research—The Netherlands (NWO-VIDI 864.12.007; H.M.); Van Gogh programme grant (VGP.17/13; A.J. and H. M.) and Imperial College London—U.K. (V.A.).

## Author contributions

V.A. and A.J. conceived and supervised the project, and together wrote the manuscript. E.B., E.C., W.M., L.J., R.A.T., H.M. and H.G.S. generated, processed BS-seq, ChIP-seq and CHi-C datasets, and/or performed bioinformatic analyses. E.B., V.B., R.A.T., K.H.T.M., Y.A. and R.A.d.S. performed experimental validation.

## Competing interests

The authors declare no competing interests.
