## [Peer Review File · Nature Communications]

Reviewers' comments:

Reviewer #1 (Remarks to the Author):

In this manuscript, Bell et al. analyse DNA methylation changes at super-enhancers during the transition from murine naïve to primed pluripotency. The authors move from the observation that super-enhancers can be divided into sub-regions displaying low methylation, linked together by stretches of methylated DNA. They show how methylation valleys in super enhancer regions correspond to sites of TF binding that preferentially interact with promoters, and argue that these regions show independent fate during the transition from naïve to primed pluripotency. They identify two classes of methylated sub-regions: one remains unmethylated in primed cells (PU), while the second gains DNA methylation (DM). Gain of methylation is linked to inactivation of DM enhancer sub-regions, as shown by the downregulation of putative target genes. The authors argue that DM regions represent particularly dynamic regulatory regions that show metastable DNA methylation profiles due to the ability to concomitantly recruit de-novo methyltransferases and TET enzymes. Such dynamics are highlighted by the heterogeneous methylation of DM regions in cells kept in culture conditions that allow spontaneous differentiation. The author propose that DM regions are particularly dependent on the activity of transcription factors characteristically expressed in the naïve state, and pinpoint *Esrrb* as the main regulator of these elements. The last section of the study shows how loss of *Esrrb*, while having little effect on histone acetylation, specifically impairs recruitment of Mediator at DM regions, in this way affecting expression of putative target genes.

In its first part, this study does not report results that are novel pre-se. Yet, existing data are re-analysed with an original perspective and provide interesting insights into the interplay between TF binding and DNA methylation dynamics during enhancer decommissioning.

For instance, super-enhancers are known to be formed by the union of independent modules, or constituent enhancers [1-6]. The finding that single enhancers in broader clusters interact preferentially with promoter and direct the topological organisation of regulatory units has been already discussed, leading to the proposal of the concept of hub enhancers [7]. More specifically, the existence of complex methylation profiles at super-enhancers has been reported, and linked to TF binding [8-10]. Similarly, the changes in methylation at regulatory regions during differentiation have been extensively analysed [9-11]. Finally, several studies have characterised the changes in enhancer activity and transcription factor binding during the conversion between naïve and primed pluripotency, identifying the major players in this transition (for the first report see [12]).

The merit of this work is to bring together existing notions, and integrate previous data with new results, to understand in detail how transcription factors shape the activity and the chromatin configuration of broad regulatory regions. In this respect, this study, through a comprehensive and rigorous analysis of previously fragmentary information, provides the first solid evidence that individual units of super-enhancer regions display divergent behaviour during cell fate transitions. This constitutes a significant advance over the exiting literature. Preliminary evidence that the recruitment of Tet and Dnmt proteins at individual enhancers might affect the dynamics of decommissioning is also provided.

In the second part of the manuscript, building on the current knowledge of the mode of action of nuclear receptors and their previous work, the authors characterise the genome-wide effects of loss of *Esrrb* on the recruitment of components of the basal transcriptional machinery and the chromatin state of enhancer regions. The data presented show strong consequences after *Esrrb* depletion, and indicates how nuclear receptor binding can modulate enhancer function independently of the deposition of active histone marks, by specifically controlling recruitment of the mediator complex. This genome-wide characterisation of the effect of *Esrrb* depletion in ESCs is novel, and extends our current understanding of the function of this transcription factor, in particular at super-enhancer regions.

Major concerns:

1) The causal connection between transcription factor binding, Dnmts and Tet recruitment, and DNA methylation changes at DM and PU regions is not explored. The nature of such dependencies

is central to the conclusions of this work. For instance, transcription factors might be directly recruiting Dnmt and Tet activities at these regions, as previously described. This could be easily addressed at individual regions in depletion experiments. Conversely, DNA methylation might affect transcription factor binding. The literature exploring these reciprocal influences should be discussed.

2) It should be consistently made clear when datasets from published studies are used, possibly providing an adequate reference in the figure legends. In some instances it is difficult to understand which datasets are being analysed. In this regard, at Line 129 reference [9] is cited as a source of data for methylation profiling in EpiSC. Is this correct?

3) The authors should explore more directly how their description of the methylation changes occurring at individual super-enhancer modules relates to the findings of studies characterising enhancer decommissioning during the transition from naïve to primed pluripotency. The regions here defined as super-enhancers are broad, and therefore the DM and PU sub-regions identified in this study likely coincide with single enhancers regions defined by transcription factor binding in previous reports. It would be interesting to know to what extent the conclusion of previous reports are relevant to understand the control of super-enhancer function during loss of naïve pluripotency.

General comment:

The description of the widespread loss of Mediator occupancy at regulatory regions in *Esrrb* null cells is a major contribution of this work and is a result of broad relevance. It would be important for the authors to extend the description of the molecular consequences of these events. In particular, describing the effects of loss of Mediator on RNAPolII occupancy at distal elements, eRNA production, and chromatin topology could be particularly informative. Mediator has been proposed to facilitate the formation of enhancer-promoter contacts, in a process that involves Cohesin, CTCF and the interaction with eRNAs. These events seem to play a particularly important role in mediating transcriptional activation by nuclear receptors [13-16]. The selective loss of mediator at a subset of enhancer regions in response to *Esrrb* depletion provides an ideal model to establish a yet elusive mechanistic connection between TF binding, the recruitment of coactivators and the basal transcriptional machinery, and the formation of functionally relevant enhancer-promoter interactions. Any additional effort in this direction would greatly add to the significance of this study.

Minor remarks:

Line 82: Fig S1a and b show that SE display a lower dynamic range of GC and CpG content compared to TE, and not the opposite.

Fig S1c: Similar data should be presented for TE in ESCs.

Line 116 and Line 210: Accessibility and DNA methylation should not be used as interchangeable terms.

Line 171: The call to fig S5 presenting RT-qPCR results and not to Fig 3c showing RNA-seq analysis is confusing.

Line 186: In Figure 3f the label PM should be probably replaced with INT.

Line 197: The authors make the point that methylation levels at DM regions show fluctuations in cells grown in serum. For this, compatibly with technical limitations, the distribution of methylation levels at single enhancers and in single cells should be shown (for example as heatmaps of methylation at single alleles), rather than presenting the average of all elements in PU, DM and INT classes (as in Figure 4a) or the average methylation levels at single loci (as in Figure 4b).

Although methylation changes seem to be coherently occurring at all enhancers in cells exiting naive pluripotency [17], averaging all regions in a class might mask substantial levels of variation.

Line 227: There is no direct evidence of oscillatory binding of Tet and Dnmts proteins at enhancers. This sentence should be rephrased.

Line 315: The conclusions of a previous study dissecting the activity of pluripotency transcription factors at the *Klf4* enhancer [18] should be mentioned in relation to the results presented.

Line 342: Cite some of the numerous studies analysing the relative functional importance of constituent enhancers in the context of broader super-enhancers.

1. Pott, S. and J.D. Lieb, What are super-enhancers? *Nat Genet*, 2015. 47(1): p. 8-12.

2. Moorthy, S.D., et al., Enhancers and super-enhancers have an equivalent regulatory role in embryonic stem cells through regulation of single or multiple genes. *Genome Res*, 2017. 27(2): p. 246-258.
3. Hay, D., et al., Genetic dissection of the alpha-globin super-enhancer in vivo. *Nat Genet*, 2016. 48(8): p. 895-903.
4. Shin, H.Y., et al., Hierarchy within the mammary STAT5-driven Wap super-enhancer. *Nat Genet*, 2016. 48(8): p. 904-911.
5. Barakat, T.S., et al., Functional Dissection of the Enhancer Repertoire in Human Embryonic Stem Cells. *Cell Stem Cell*, 2018. 23(2): p. 276-288 e8.
6. Xie, S., et al., Multiplexed Engineering and Analysis of Combinatorial Enhancer Activity in Single Cells. *Mol Cell*, 2017. 66(2): p. 285-299 e5.
7. Huang, J., et al., Dissecting super-enhancer hierarchy based on chromatin interactions. *Nat Commun*, 2018. 9(1): p. 943.
8. Heyn, H., et al., Epigenomic analysis detects aberrant super-enhancer DNA methylation in human cancer. *Genome Biol*, 2016. 17: p. 11.
9. Stadler, M.B., et al., DNA-binding factors shape the mouse methylome at distal regulatory regions. *Nature*, 2011. 480(7378): p. 490-5.
10. Xie, W., et al., Epigenomic analysis of multilineage differentiation of human embryonic stem cells. *Cell*, 2013. 153(5): p. 1134-48.
11. Gifford, C.A., et al., Transcriptional and epigenetic dynamics during specification of human embryonic stem cells. *Cell*, 2013. 153(5): p. 1149-63.
12. Buecker, C., et al., Reorganization of enhancer patterns in transition from naive to primed pluripotency. *Cell Stem Cell*, 2014. 14(6): p. 838-53.
13. Kagey, M.H., et al., Mediator and cohesin connect gene expression and chromatin architecture. *Nature*, 2010. 467(7314): p. 430-5.
14. Li, W., et al., Functional roles of enhancer RNAs for oestrogen-dependent transcriptional activation. *Nature*, 2013. 498(7455): p. 516-20.
15. Lai, F., et al., Activating RNAs associate with Mediator to enhance chromatin architecture and transcription. *Nature*, 2013. 494(7438): p. 497-501.
16. Hsieh, C.L., et al., Enhancer RNAs participate in androgen receptor-driven looping that selectively enhances gene activation. *Proc Natl Acad Sci U S A*, 2014. 111(20): p. 7319-24.
17. Rulands, S., et al., Genome-Scale Oscillations in DNA Methylation during Exit from Pluripotency. *Cell Syst*, 2018. 7(1): p. 63-76 e12.
18. Xie, L., et al., A dynamic interplay of enhancer elements regulates Klf4 expression in naive pluripotency. *Genes Dev*, 2017. 31(17): p. 1795-1808.

Reviewer #2 (Remarks to the Author):

In this work, Bell and colleagues performed an in-depth analysis of CpG methylation patterns at Super Enhancers (SEs) regions in naive (ESCs) and primed embryonic stem cells (EpiSCs). In ESCs, within SEs, the authors found that some regions can be unmethylated and interact with active promoters as neighboring methylated regions interact with such promoter less frequently. Further characterization of methylation behaviors in ESC and in primed EpiSCs led the authors to define subregions in SEs with differentially methylated (DM), highly methylated (INT), or unmethylated (PU) characteristics. Moreover, the authors define two classes of SEs: a class I that remains unmethylated in ESC and EpiSCs and is associated to genes which expression is maintained during differentiation. A class II which contain at least one differentially methylated region and associate with genes displaying a reduction in expression during differentiation. The authors postulate that DM regions within class II SEs can be found in a metastable methylation state, poising the SEs for decommissioning during differentiation. The authors also propose a mechanism based on several transcription factors which expression associate with differences in naïve and prime states. In particular, they identified ESRRB as a potent candidate regulating DM

regions activities during the pluripotent to naïve cell transition. The authors show that in ESCs depleted for ESRRB, the class II SE interacting genes are less expressed, suggesting an active role for ESRRB in priming cell for differentiation, and loss of pluripotency. Finally, the authors propose a mechanism by which ESRRB1 recruit Mediator and that ESRRB1 loss upon differentiation would de-activate SEs DM subregions leading to a decreased expression of associated genes.

Generally, this paper contains novel insights into the function of super enhancers and the transition between cellular states. In principle, the idea of a metastable enhancer state poising the exit of pluripotency is supported by the presented data and exciting. However, the mechanistical insights shown by the authors are not very convincing and are oriented toward a Mediator-effect and not a CpG methylation induced effect. Moreover, the paper is difficult to follow and the figures could generally be improved.

Major comments

1. PU regions in Fig. 2a appear to be at the edge of the featured super enhancers, is that a systematic property of this region and if yes, could the author comment on it?
2. The authors state that DM regions becomes gradually decommissioned as primed pluripotency is established in vivo. However, they only show 2 stages and thus progressivity of the methylation is impossible to assess. An extra stage should be added, as for RNA-seq or the authors should correct the statement in the text.
3. The authors do not comment on the fact that INT regions are not at all methylated in E3.5 ICM. In fact they behave like DM regions in vivo and not like INT regions. It is necessary to discuss this discrepancy between in vitro and in vivo model system.
4. If some SEs bear DM and PU regions, does it mean that SE are module of completely independent enhancers, that target genes independently as well? In that case, what is the role/function of SEs? In fact, it appears contradictory to this reviewer that SEs, originally defined as a synergistic assembly of enhancer activities achieving strong coordinated regulation are here characterized as an assembly of independently acting regulatory units. In fact, it seems to this reviewer that the here-presented data argue for a non-synergistic effect of the elements composing SEs, an observation that should be discussed by the authors.
5. Promoters silenced in EpiSCs were found associated with DM regions as active ones were associated with region containing at least one PU (lanes 149-156). Both associations are not mutually exclusive and it would be interesting to see the association between silenced promoters and regions containing at least one PU. In reverse it would be of interest to see the association between persistently active promoters and DM regions
6. As PU regions are systematically enriched for the enhancer mark H3K27ac, it would be of interest to know how DM regions behave with respect to this mark. From the data presented in in fig 3e,f it seems that there is no enrichment neither depletion.
7. From what mechanism originates the expression variation in genes, such as *Esrrb*, that create the postulated metastable state. Is their own enhancer in a metastable state? And if yes by what mechanism? This authors should discuss this.
8. This reviewer has several reserve concerning the molecular mechanism proposed by the authors. In particular, this reviewer feels that the quality of chip experiments in figure 6 is not optimal. The ChIP-qPCR in figure 6d shows only very limited enrichment over background, it is thus very difficult to know if indeed a real loss of Med1 occurs at the different regions. Moreover, in figure 6f, some regions appear to loose MED1 binding in KO although they are not bound by ESRRB in wildtype ESCs. The authors should provide a quality assessment of their data and a thorough comparison of mediator binding at regions that are not bound by ESRRB. This is important because OCT4 was recently shown to recruit MED1 (Boija et al., 2018 Cell) and in the current work, the binding of OCT4 is unchanged in ESSRB KO cells.
9. The loss of mediator provides an ad hoc explanation of the ESRRB-induced expression loss but has little to do with DNA methylation and the metastable state proposed by the authors. This reviewer recommends the authors to produce methylation data in ESRRB KO ESCs in order to link changes in ESRRB and methylation at SEs.

Minor comments

1. To determine how SEs subregions interact with promoters, the authors used a capture Hi-C dataset based on HindIII. This enzyme is a 6 bp cutter, and thereby cuts in average every 4kb, it is thus difficult to imagine how they could achieve a coverage good enough to make the difference between methylated and unmethylated subregions. Could the author comment on that?
2. The authors use the term "accessible" to define unmethylated regions (see for instance lane 116). This reviewer is not sure if this is an accepted terminology.
3. The authors should mention in the text how many SEs are studied in this work and how many regions are classified as PU, DM or INT.
4. It is unclear to this reviewer why the enrichment of H3K27ac is different between figures 3e,f and 4c-5e.
5. Why is the AF-2 mutant ESRRB affecting both class I and II genes set? This reviewer missed a discussion about this aspect.

Reviewer #3 (Remarks to the Author):

Bell et al. have identified that SEs can be subdivided in different regions with different methylation and TF binding properties. Interestingly, some of these regions are variably methylated and have high levels of binding of ESRRB while constant low methylated regions seem to be the main target regions for classical pluripotency factors. Overall this is a well performed study with interesting novel observations. My main concern is, that the observed differences attributed to methylation are a consequence of CpG density and less of methylation itself. This should be better discussed in the text.

Comments:

Figure 1a) I think the representation of mCpG values as -1 to 1 a bit confusion. Could this not be shown as 0 – 1 which would be the normal expected range?

Is the difference in CpG density significant – it looks very minor and may be consequence of the relatively small size of SEs?

Figure S1C: it would be nice to show the 5mC levels at proB SE and ESC SE in both conditions, ie proB SE in proB and also ESC

Figure 1B: Many TFs are known to bind CpG rich motifs and in 1A the authors show that SE have mostly unmethylated CpGs -> is the result in 1B not just a consequence of this, ie are methylation levels relevant or just CpG position?

Figure 1C: see above – is it a consequence of methylation levels?

Line 116 "ESC SE-interacting gene promoters remained largely accessible" – this should say "largely hypomethylated"

Figure 2,3: Interesting and convincing observation

Figure 3e.f: This result seems expected based on the results shown before assuming that methylation levels are related to the factors measured here. The authors could briefly discuss/mention this.

Figure 4: Could the authors have predicted the PU and DM regions by looking at DNMT3a/b and

TET1 binding profiles, ie how well would these overlap?

Figure 5C: Does ESRRB/NR5A2 show a strong variable expression in serum/LIF cells, ie does ESRRB/NR5A2 expression correlate with the status of DM in single cell datasets?

Figure 7: I am not sure if the model should suggest a different binding kinetic for each SE region or whether the differences are more between cells. This is something that the authors could discuss.

Responses to reviewers_ NCOMMS-19-08409

Reviewers' comments:

Reviewer #1 (Remarks to the Author):

In this manuscript, Bell et al. analyse DNA methylation changes at super-enhancers during the transition from murine naïve to primed pluripotency. The authors move from the observation that super-enhancers can be divided into sub-regions displaying low methylation, linked together by stretches of methylated DNA. They show how methylation valleys in super enhancer regions correspond to sites of TF binding that preferentially interact with promoters, and argue that these regions show independent fate during the transition from naïve to primed pluripotency. They identify two classes of methylated sub-regions: one remains unmethylated in primed cells (PU), while the second gains DNA methylation (DM). Gain of methylation is linked to inactivation of DM enhancer sub-regions, as shown by the downregulation of putative target genes. The authors argue that DM regions represent particularly dynamic regulatory regions that show metastable DNA methylation profiles due to the ability to concomitantly recruit de-novo methyltransferases and TET enzymes. Such dynamics are highlighted by the heterogeneous methylation of DM regions in cells kept in culture conditions that allow spontaneous differentiation. The author propose that DM regions are particularly dependent on the activity of transcription factors characteristically expressed in the naïve state, and pinpoint Esrrb as the main regulator of these elements. The last section of the study shows how loss of Esrrb, while having little effect on histone acetylation, specifically impairs recruitment of Mediator at DM regions, in this way affecting expression of putative target genes. In its first part, this study does not report results that are novel pre-se. Yet, existing data are re-analysed with an original perspective and provide interesting insights into the interplay between TF binding and DNA methylation dynamics during enhancer decommissioning. For instance, super-enhancers are known to be formed by the union of independent modules, or constituent enhancers [1-6]. The finding that single enhancers in broader clusters interact preferentially with promoter and direct the topological organisation of regulatory units has been already discussed, leading to the proposal of the concept of hub enhancers [7]. More specifically, the existence of complex methylation profiles at super-enhancers has been reported, and linked to TF binding [8-10]. Similarly, the changes in methylation at regulatory regions during differentiation have been extensively analysed [9-11]. Finally, several studies have characterised the changes in enhancer activity and transcription factor binding during the conversion between naïve and primed pluripotency, identifying the major players in this transition (for the first report see [12]). The merit of this work is to bring together existing notions, and integrate previous data with new results, to understand in detail how transcription factors shape the activity and the chromatin configuration of broad regulatory regions. In this respect, this study, through a comprehensive and rigorous analysis of previously fragmentary information, provides the first solid evidence that individual units of super-enhancer regions display divergent behaviour during cell fate transitions. This constitutes a significant advance over the exiting literature. Preliminary evidence that the recruitment of Tet and Dnmt proteins at individual enhancers might affect the dynamics of decommissioning is also provided. In the second part of the manuscript, building on the current knowledge of the mode of action of nuclear receptors and their previous work, the authors characterise the genome-wide effects of loss of Esrrb on the recruitment of components of the basal transcriptional machinery and the chromatin state of enhancer regions. The data presented show strong consequences after Esrrb depletion, and indicates how nuclear receptor binding can modulate enhancer function independently of the deposition of active histone marks, by specifically controlling recruitment of the mediator complex. This genome-wide characterisation of the effect of Esrrb depletion in ESCs is novel, and extends our current understanding of the function of this transcription factor, in particular at super-enhancer regions.

We thank the reviewer for their enthusiasm about our paper and for their compliments on our “comprehensive and rigorous analysis”.

Major concerns:

1) The causal connection between transcription factor binding, Dnmts and Tet recruitment, and DNA methylation changes at DM and PU regions is not explored. The nature of such dependencies is central to the conclusions of this work. For instance, transcription factors might be directly recruiting Dnmt and Tet activities at these regions, as previously described. This could be easily addressed at individual regions in depletion experiments. Conversely, DNA methylation might affect transcription factor binding. The literature exploring these reciprocal influences should be discussed.

Next to exploring the literature as suggested by the reviewer (see below), we performed several analyses to gain further insights into this issue:

- We have now examined the impact of ESRRB depletion on CpG methylation along *Klf4*-associated super-enhancer (SE) used in this study as a model locus (see new Figure 5g). We show that the loss of ESRRB binding promotes *de novo* methylation at DM subregions in *Esrrb*^{-/-} ESCs. The observed increase in CpG methylation is consistent with the destabilisation of *Klf4* expression in these cells (Supplementary Figure 7g). As a control, we confirm that *Klf4*-associated SE becomes hypermethylated in converted EpiSCs where the expression of *Klf4* and *Esrrb* is completely silenced (Figure 5g; Supplementary Figure 7h)(see also in the text line 309). Together, in terms of hierarchy, our results show that the gain in CpG methylation at DM subregions is a direct consequence of the loss of ESRRB binding.
- An interesting finding arising from our study is the differential levels of CpG methylation variance observed at DM versus PU subregions in ESCs-serum. Our initial analyses pointed to (1) DNMT3s and TETs, and (2) ESRRB as potential drivers of CpG methylation dynamics at SEs. We have now further explored these observations as follows:
 - We previously identified that differential ChIP enrichment for DNMT3A/B and TET1 might be predictive features of PU and DM subregions as now confirmed using logistic regression models (see response to reviewer 3 and text line 220). To further dissect the influence of DNMT3s and TETs binding on methylation variance, we used available single-cell BS-seq data collected from wild-type, *Dnmt3a/b* DKO and *Tet1-3* TKO ESCs (Angermulleler *et al.* Nature Methods 2016; Rulands *et al.* Cell Systems 2018). Our analyses reveal that cell-to-cell CpG methylation heterogeneity is significantly higher at DM compared to PU subregions and furthermore is dependent on DNMT3s rather than TETs activities (see new Figure 4e)(see text line 231), addressing the reviewer’s question on how DNMTs and TETs binding might influence CpG methylation at SE subregions.
 - To explore additional regulators of CpG methylation at DM regions besides DNMT3s activities, we previously interrogated parallel single-cell BS-seq and RNA-seq datasets in ESCs (Angermulleler *et al.* Nature Methods 2016) to ask whether heterogenous expression of differentiation versus pluripotency-associated genes might contribute to CpG methylation dynamics at SEs. Our

results identified ESRRB as a top candidate in contrast to other pluripotency factors including NR5A2 (see new Figure 4f). We now provide additional experimental evidence suggesting that ESRRB binding counteracts *de novo* methylation at DM subregions (see above). This, together with the ability of ESRRB to access ERRE binding sites within methylated regions (Adachi *et al.* Cell Stem Cell 2018), lead us to propose that a balance between DNMT3s and ESRRB activities instigates a metastable methylation state at DM subregions poisoning the exit of naïve pluripotency. It remains to understand how ESRRB counteracts CpG methylation upon binding. As TET binding is not a main driver of CpG methylation dynamics at SEs (as also suggested by a complementary study published during this revision period; Song *et al.* Mol Cell, Sept 2019), we did not further investigate a possible mechanism involving ESRRB-mediated recruitment of TETs. This was also motivated by the lack of current evidence for ESRRB and TET protein interactions in ESCs (Van Den Berg *et al.* Cell Stem Cell 2010; Yaser Atlasi, personal communications) as opposed, for example, to Mediator and RNA polymerase II complexes (see below).

- In addition to our novel analyses, we are now more extensively citing previous studies exploring the reciprocal influences of transcription factor binding and CpG methylation in our revised manuscript (see for example, line 369, 375 or 431).

2) It should be consistently made clear when datasets from published studies are used, possibly providing an adequate reference in the figure legends. In some instances, it is difficult to understand which datasets are being analysed. In this regard, at Line 129 reference [9] is cited as a source of data for methylation profiling in EpiSC. Is this correct?

- We have included the GSE numbers of our newly generated data (BS-seq, ChIP-seq and 4C-seq) in Methods (line 447). These data are also highlighted in the text with the mention “this study” (see for example line 114). As for Figure 2c,d and Supplementary Figure 4d,e that use different datasets for ESCs, EpiLCS and EpiSCs including our BS-seq data, we are citing the references of published datasets in figure legends and/or in the main text (see for example line 80, 127, 129 or 135). Please note that all publicly available datasets used in our study are summarised in the Supplementary Table 1 as referred to throughout our manuscript (see for example line 91).

3) The authors should explore more directly how their description of the methylation changes occurring at individual super-enhancer modules relates to the findings of studies characterising enhancer decommissioning during the transition from naïve to primed pluripotency. The regions here defined as super-enhancers are broad, and therefore the DM and PU sub-regions identified in this study likely coincide with single enhancers regions defined by transcription factor binding in previous reports. It would be interesting to know to what extent the conclusion of previous reports are relevant to understand the control of super-enhancer function during loss of naïve pluripotency.

There have been 4 important findings in previous studies that our paper further refines:

- Previous works on enhancer decommissioning report a gradual loss of the enhancer markers H3K27ac, OCT4 and ATAC-seq signals at naïve enhancers during the ESC-to-EpiSC transition (Buecker *et al.* Cell Stem Cell 2014). We find that the decommissioned DM subregions within SEs follow a similar path than single naïve enhancer units. In contrast, we find that PU subregions maintained an active enhancer status, indicative of their retained activity in the

primed cells. While this information was “scattered” in different figures of our initial submission, it is now gathered in the new Figure 3e (see also line 179 of our revised manuscript).

- Evidence that a core set of genes remain expressed through the transition from naïve-to-primed pluripotency was previously reported. These genes were proposed to be regulated by distinct state-specific enhancers in ESCs and EpiSCs (Factor *et al.* Cell Stem Cell 2014). In our study, we propose that a subset of enhancer units (PU) within ESC SEs continue to potentiate the expression of some SE-associated genes throughout this transition. Though our findings do not exclude that the same genes might also be regulated by state-specific enhancers, they offer novel mechanistic insights into how the pluripotency transcriptional programme might be partially reset between ESCs and EpiSCs, as further established in the developing epiblast *in vivo*. This point is now highlighted in our revised text line 379.
- Additionally, we show that the functional partitioning of enhancer units might not be restricted to SEs but also applies to isolated TE enhancers. Indeed, we find that TEs similarly divide into DM-like and PU-like subtypes, with DM-like TE regions being hypermethylated in EpiSCs and selectively regulated by ESRRB in ESCs (new Supplementary Figure 9) as discussed in the text line 437. This highlights a pivotal role for ESRRB in partitioning and regulating both naïve single and clustered enhancers at the exit of ESC pluripotency.
- Finally, we discuss the possibility that PU subregions might be transiently protected from *de novo* methylation by specific transcription factors whose expression is regionalised in the patterning epiblast (e.g. FOXD3 or GRHL2)(Respuela *et al.* Cell Stem Cell 2016; Chen *et al.* Cell Stem Cell 2018). These regions might subsequently acquire CpG methylation upon differentiation in a lineage-specific manner as examined in haematopoietic cells in our revised manuscript (see new Supplementary Figure 4g and in the text line 394).

General comment:

The description of the widespread loss of Mediator occupancy at regulatory regions in *Esrrb* null cells is a major contribution of this work and is a result of broad relevance. It would be important for the authors to extend the description of the molecular consequences of these events. In particular, describing the effects of loss of Mediator on RNAPolII occupancy at distal elements, eRNA production, and chromatin topology could be particularly informative. Mediator has been proposed to facilitate the formation of enhancer-promoter contacts, in a process that involves Cohesin, CTCF and the interaction with eRNAs. These events seems to play a particularly important role in mediating transcriptional activation by nuclear receptors [13-16]. The selective loss of mediator at a subset of enhancer regions in response to *Esrrb* depletion provides an ideal model to establish a yet elusive mechanistic connection between TF binding, the recruitment of coactivators and the basal transcriptional machinery, and the formation of functionally relevant enhancer-promoter interactions. Any additional effort in this direction would greatly add to the significance of this study.

- We agree with the reviewer and have now significantly extended the description of the molecular consequences of ESRRB depletion at the *Klf4* locus by performing additional experiments as follows.
 - Analysis of RNA polymerase II (POL2) recruitment and transcription of eRNA along the *Klf4*-associated SE in *Esrrb* *-/-* and control ESCs, showing a significant loss of POL2 recruitment and eRNA expression corroborating with the loss of MED1 occupancy. The results are presented in the new Figures 6c,d and

described in the text lines 334 (see also new Supplementary Figure 8e,f for an extended analysis).

- Analysis of chromatin interactions using 4C-seq assays in *Esrrb* ^{-/-} and control ESC- serum together with ESC-2i used as control. The results are presented in the new Figure 6b and show the gradual loss of promoter-SE interactions from 2i to serum and upon ESRRB depletion in ESCs (see also text line 327).
- Analysis of the CpG methylation profile of *Klf4*-associated SE in *Esrrb* ^{-/-} and control ESCs as well as in converted EpiSCs (see above).

Together, these analyses clearly show the pivotal role played by ESRRB in regulating the enhancer activity of DM subregions within SEs, and furthermore establish a mechanistic link between nuclear receptor binding, the recruitment of co-activators and the basal transcriptional machinery, and the formation of functionally relevant enhancer-promoter interactions in ESCs.

Minor remarks:

Line 82: Fig S1a and b show that SE display a lower dynamic range of GC and CpG content compared to TE, and not the opposite. Fig S1c: Similar data should be presented for TE in ESCs.

- We agree with the reviewer that SEs display lower variance in size, GC and CpG content than TEs; however, this might merely reflect the higher number of TEs examined compared to SEs in this analysis (Supplementary Figure 1a). Supplementary Figure 1b and new Supplementary Figure 1c now include the analysis of CpG methylation at ESC TEs along with ESC and proB cell SEs in ESCs and in B cells, respectively. The results highlight differences in the methylation profiles of SEs and TEs with ESC and proB cell SEs showing similar features in a lineage-specific manner as highlighted in the text line 86.

Line 116 and Line 210: Accessibility and DNA methylation should not be use as interchangeable terms.

- We thank the reviewer for pointing this out; this has been corrected as appropriate.

Line 171: The call to fig S5 presenting RT-qPCR results and not to Fig 3c showing RNA-seq analysis is confusing.

- The order of this result section has now been changed as suggested by the reviewer (see text line 161). Please note that figure 3c (now figure 3d) does not present RNA-seq data but RT-qPCR data obtained during *in vitro* conversion of ESCs into EpiSCs, as also shown in Supplementary figure 5a-c.

Line 186: In Figure 3f the label PM should be probably replaced with INT.

- This figure panel has been corrected (now shown as Supplementary Figure 5e).

Line 197: The authors make the point that methylation levels at DM regions show fluctuations in cells grown in serum. For this, compatibly with technical limitations, the distribution of methylation levels at single enhancers and in single cells should be shown (for example as heatmaps of methylation at single alleles), rather than presenting the average of all elements in PU, DM and INT

classes (as in Figure 4a) or the average methylation levels at single loci (as in Figure 4b). Although methylation changes seem to be coherently occurring at all enhancers in cells exiting naive pluripotency [17], averaging all regions in a class might mask substantial levels of variation.

- Unfortunately, the coverage of the sc-BS-seq dataset used is not sufficient to examine the distribution of methylation levels at single SE subregions and in single cells as suggested; most subregions have too few mapped reads to make a reliable methylation call for a significant proportion of individual cells. We have, however, included in our revised manuscript alternative data presentations and analyses alongside Figure 4a and Figure 4b (notably, the original Figure 4b is labelled Figure 4c in the revised manuscript) as follows:
 - To illustrate the distributions of methylation levels across SE subregions, 5 individual cells were randomly selected from each ESC cluster (i.e. 2i ESCs, serum naïve-like and primed-like ESCs). For each of the selected cells, the average methylation across each individual SE subregion was computed and these distributions for each cell were illustrated with violin plots (distributions for PU, DM and INT subregions shown separately). Cells were ordered in each plot by their average methylation level across all subregions of the corresponding type (see new Figure 4b).
 - For each ESC cluster, the variance in CpG methylation levels across SE subregions of a specific type (i.e. PU or DM) was computed for each cell belonging to the corresponding cluster. The distribution of these methylation variances for each cluster, for each subregion type is shown through a box plot (see new Supplementary Figure 6a).

We agree with the reviewer that there is indeed variation at the level of individual SE subregions, but these analyses reveal that the levels of variation at the DM subregions are distinct between ESCs of the different clusters, reinforcing our observation that the methylation of DM subregions seems intrinsically linked to the exit of naïve pluripotency.

Line 227: There is no direct evidence of oscillatory binding of Tet and Dnmts proteins at enhancers. This sentence should be rephrased.

- Indeed, the reviewer is correct. This sentence has been removed from our manuscript, and Figure 7 (model) has been edited accordingly.

Line 315: The conclusions of a previous study dissecting the activity of pluripotency transcription factors at the *Klf4* enhancer [18] should be mentioned in relation to the results presented.

- This study is now cited in our result section in relation with the retention of OCT4 and P300 binding upon ESRRB depletion in ESCs at the *Klf4* locus (see text line 326).

Line 342: Cite some of the numerous studies analysing the relative functional importance of constituent enhancers in the context of broader super-enhancers.

- We are now citing a number of these previous studies in our discussion as suggested (see for example line 364, 369 or 372).

1. Pott, S. and J.D. Lieb, What are super-enhancers? Nat Genet, 2015. 47(1): p. 8-12.

2. Moorthy, S.D., et al., Enhancers and super-enhancers have an equivalent regulatory role in embryonic stem cells through regulation of single or multiple genes. *Genome Res*, 2017. 27(2): p. 246-258.
3. Hay, D., et al., Genetic dissection of the alpha-globin super-enhancer in vivo. *Nat Genet*, 2016. 48(8): p. 895-903.
4. Shin, H.Y., et al., Hierarchy within the mammary STAT5-driven Wap super-enhancer. *Nat Genet*, 2016. 48(8): p. 904-911.
5. Barakat, T.S., et al., Functional Dissection of the Enhancer Repertoire in Human Embryonic Stem Cells. *Cell Stem Cell*, 2018. 23(2): p. 276-288 e8.
6. Xie, S., et al., Multiplexed Engineering and Analysis of Combinatorial Enhancer Activity in Single Cells. *Mol Cell*, 2017. 66(2): p. 285-299 e5.
7. Huang, J., et al., Dissecting super-enhancer hierarchy based on chromatin interactions. *Nat Commun*, 2018. 9(1): p. 943.
8. Heyn, H., et al., Epigenomic analysis detects aberrant super-enhancer DNA methylation in human cancer. *Genome Biol*, 2016. 17: p. 11.
9. Stadler, M.B., et al., DNA-binding factors shape the mouse methylome at distal regulatory regions. *Nature*, 2011. 480(7378): p. 490-5.
10. Xie, W., et al., Epigenomic analysis of multilineage differentiation of human embryonic stem cells. *Cell*, 2013. 153(5): p. 1134-48.
11. Gifford, C.A., et al., Transcriptional and epigenetic dynamics during specification of human embryonic stem cells. *Cell*, 2013. 153(5): p. 1149-63.
12. Buecker, C., et al., Reorganization of enhancer patterns in transition from naive to primed pluripotency. *Cell Stem Cell*, 2014. 14(6): p. 838-53.
13. Kagey, M.H., et al., Mediator and cohesin connect gene expression and chromatin architecture. *Nature*, 2010. 467(7314): p. 430-5.
14. Li, W., et al., Functional roles of enhancer RNAs for oestrogen-dependent transcriptional activation. *Nature*, 2013. 498(7455): p. 516-20.
15. Lai, F., et al., Activating RNAs associate with Mediator to enhance chromatin architecture and transcription. *Nature*, 2013. 494(7438): p. 497-501.
16. Hsieh, C.L., et al., Enhancer RNAs participate in androgen receptor-driven looping that selectively enhances gene activation. *Proc Natl Acad Sci U S A*, 2014. 111(20): p. 7319-24.
17. Rulands, S., et al., Genome-Scale Oscillations in DNA Methylation during Exit from Pluripotency. *Cell Syst*, 2018. 7(1): p. 63-76 e12.
18. Xie, L., et al., A dynamic interplay of enhancer elements regulates Klf4 expression in naive pluripotency. *Genes Dev*, 2017. 31(17): p. 1795-1808.

Reviewer #2 (Remarks to the Author):

In this work, Bell and colleagues performed an in-depth analysis of CpG methylation patterns at Super Enhancers (SEs) regions in naive (ESCs) and primed embryonic stem cells (EpiSCs). In ESCs, within SEs, the authors found that some regions can be unmethylated and interact with active promoters as neighboring methylated regions interact with such promoter less frequently. Further characterization of methylation behaviors in ESC and in primed EpiSCs led the authors to define subregions in SEs with differentially methylated (DM), highly methylated (INT), or unmethylated (PU) characteristics. Moreover, the authors define two classes of SEs: a class I that remains unmethylated in ESC and EpiSCs and is associated to genes which expression is maintained during differentiation. A class II which contain at least one differentially methylated region and associate with genes displaying a reduction in expression during differentiation. The authors postulate that DM regions within class II SEs can be found in a metastable methylation state, poisoning the SEs for decommissioning during differentiation. The authors also propose a mechanism based on several transcription factors which expression associate with differences in naïve and prime states. In

particular, they identified ESRRB as a potent candidate regulating DM regions activities during the pluripotent to naïve cell transition. The authors show that in ESCs depleted for ESRRB, the class II SE interacting genes are less expressed, suggesting an active role for ESRRB in priming cell for differentiation, and loss of pluripotency. Finally, the authors propose a mechanism by which ESRRB1 recruit Mediator and that ESRRB1 loss upon differentiation would de-activate SEs DM subregions leading to a decreased expression of associated genes.

Generally, this paper contains novel insights into the function of super enhancers and the transition between cellular states. In principle, the idea of a metastable enhancer state poising the exit of pluripotency is supported by the presented data and exciting. However, the mechanistical insights shown by the authors are not very convincing and are oriented toward a Mediator-effect and not a CpG methylation induced effect. Moreover, the paper is difficult to follow, and the figures could generally be improved.

We thank the reviewer for their positive evaluation of our paper. We have now provided more mechanistic insights and also improved various figures, as outlined in our responses to the reviewers.

Major comments

1. PU regions in Fig. 2a appear to be at the edge of the featured super enhancers, is that a systematic property of this region and if yes, could the author comment on it?

- We did not find any bias in the localisation of PU subregions when systematically inspecting all SEs (data not shown). An example of alternative location can be seen along *Spry4*-associated SE (see Figure 6f; right panel).

2. The authors state that DM regions becomes gradually decommissioned as primed pluripotency is established *in vivo*. However, they only show 2 stages and thus progressivity of the methylation is impossible to assess. An extra stage should be added, as for RNA-seq or the authors should correct the statement in the text.

- We are now showing *in vivo* CpG methylation profiles at PU, DM and INT subregions for additional developmental times, namely E3.5, E4, E5.5 and E6.5 (see new Figure 2d), as well as examining the impact of DM subregion decommissioning on gene expression at the same exact times (see new Figure 3c). The results support a model in which the DM subregions become gradually decommissioned during transition towards primed pluripotency (see also text line 132).

3. The authors do not comment on the fact that INT regions are not at all methylated in E3.5 ICM. In fact they behave like DM regions *in vivo* and not like INT regions. It is necessary to discuss this discrepancy between *in vitro* and *in vivo* model system.

- We agree with the reviewer that INT regions appear to be relatively poorly methylated in E3.5-E4 epiblasts compared to ESCs grown under serum/LIF conditions. However, this pattern might be best compared with the methylation profile of INT subregions in ESCs grown under 2i/LIF conditions. Indeed, these cells were proposed to most closely resemble the early (E4) epiblast *in vivo* (Boroviak *et al.* Nature Cell Biology 2014). The methylation profiles of PU, DM and INT subregions in 2i-ESCs is now shown along with serum-ESCs, EpiLCs and EpiSCs in our new Figure 2c.

4. If some SEs bear DM and PU regions, does it mean that SE are module of completely independent enhancers, that target genes independently as well? In that case, what is the role/function of SEs? In fact, it appears contradictory to this reviewer that SEs, originally defined as a synergistic assembly of enhancer activities achieving strong coordinated regulation are here characterized as an assembly of independently acting regulatory units. In fact, it seems to this reviewer that the here-presented data argue for a non-synergistic effect of the elements composing SEs, an observation that should be discussed by the authors.

- Whether enhancers within SEs are functionally interdependent units (Hnisz *et al.* Mol. Cell 2015) exhibiting synergy (Suzuki *et al.* Cell 2017) or act independently in an additive (Hay *et al.* Nature Genetics 2016) or a partially redundant manner (Moorthy *et al.* Genome Res 2017) is still a matter of debate (see text line 364). While we report that PU and DM within SEs are both active in ESCs – i.e. they are accessible and hypomethylated, highly bound by TFs and co-regulators, engaged in interactions with active promoters (see new Supplementary Figure 3i; Figure 6b) as well as transcribing eRNA (see new Supplementary Figure 8c; Figure 6c,d), our findings do not infer on whether DM and PU enhancer units act in a synergistic manner in ESCs. In contrast, we propose novel and compelling evidence that PU and DM subregions follow independent fates upon cell state transitions that can be predicted based on pre-exiting distinctive CpG methylation dynamics and chromatin configurations in ESCs as discussed in the text line 379. In this respect, our model might argue against a synergistic model at least in the primed state.

5. Promoters silenced in EpiSCs were found associated with DM regions as active ones were associated with region containing at least one PU (lanes 149-156). Both associations are not mutually exclusive, and it would be interesting to see the association between silenced promoters and regions containing at least one PU. In reverse it would be of interest to see the association between persistently active promoters and DM regions

- We agree with the reviewer that these associations might not be mutually exclusive. This can be seen in Figure 3a where not all genes (dots) fit well with the prediction, i.e. are not on the diagonal. However, important naïve genes associated with DM subregions follow the prediction and are strongly downregulated (magenta dots in Figure 3a) and some primed genes associated with SE containing at least one PU remain expressed or even upregulated (green dots). We have now better described this figure in the text line 144, highlighting that some genes do escape the prediction of our model. We would like to point out that pluripotency genes do not exclusively interact with SEs, but also with (many) TEs. Taking these TEs into account would likely improve the observed correlation in Figure 3a and might explain some of the discrepancies between PU/DM and maintenance/loss of gene expression. Our focus on CpG methylation at SEs - a small subset of all enhancers in ESCs - arguably has predictive limitations, but nevertheless shows clear and significant trends as examined both *in vitro* (Supplementary Figure 4f) and *in vivo* (Figure 3c) (see also revised text line 161).

6. As PU regions are systematically enriched for the enhancer mark H3K27ac, it would be of interest to know how DM regions behave with respect to this mark. From the data presented in in fig 3e,f it seems that there is no enrichment neither depletion.

- We have added a new figure panel allowing for direct comparison of the levels of H3K27ac enrichment at PU, DM and INT subregions in ESCs, EpiLCs and EpiSCs (see new Figure 3e).

The results show a gradual reduction in H3K27ac deposition at DM as opposed to PU subregions.

7. From what mechanism originates the expression variation in genes, such as *Esrrb*, that create the postulated metastable state. Is their own enhancer in a metastable state? And if yes by what mechanism? This authors should discuss this.

- Our results suggest that a balance between ESRRB and DNMT3 activities instigates a metastable state at Class II (DM-containing) SEs typified by a high level of CpG methylation dynamics, leading to heterogeneous transcriptional and cellular states within ESCs. This point is now further supported by the following additional experiments/computational analyses (see also response to reviewer 1, point 1):
 - We provide novel evidence indicating that ESRRB acts at DM subregions by counteracting *de novo* methylation upon binding (at least in the *Klf4*-associated SE that we used as a model; see new Figure 5g).
 - By examining the impact of DNMT3s versus TET depletion on methylation variance at the single-cell level, we show that DNMT3 activities are important drivers of CpG methylation heterogeneity at DM subregions (see new Figure 4e) as previously identified for ESRRB (see ROC analysis in Figure 4f; right panel).
 - We confirm that *Esrrb* itself is associated with a Class II (DM-containing only) SE subjected to CpG methylation oscillations (see new Figure 4c; lower panel) with impact on its own expression pattern (see new Supplementary Figure 6f) and that of other Class II SE-associated gene targets (see Supplementary Figure 6c,d & f).

Based on these findings, we propose that the dynamic expression of ESRRB in ESCs, together with its ability to access ERRE binding sites within a methylated region (Adachi *et al.* Cell Stem Cell 2018) or to inhibit DNMT3-mediated methylation upon binding (this study) favours the emergence of cell-to-cell CpG methylation heterogeneity at DM subregions. This suggests a novel mechanism by which Class II SEs might be prone for decommissioning upon exit from naïve pluripotency (as discussed in the text line 426).

8. This reviewer has several reserve concerning the molecular mechanism proposed by the authors. In particular, this reviewer feels that the quality of chip experiments in figure 6 is not optimal. The ChIP-qPCR in figure 6d shows only very limited enrichment over background, it is thus very difficult to know if indeed a real loss of Med1 occurs at the different regions. Moreover, in figure 6f, some regions appear to loose MED1 binding in KO although they are not bound by ESRRB in wildtype ESCs. The authors should provide a quality assessment of their data and a thorough comparison of mediator binding at regions that are not bound by ESRRB. This is important because OCT4 was recently shown to recruit MED1 (Boija *et al.*, 2018 Cell) and in the current work, the binding of OCT4 is unchanged in ESRRB KO cells.

- With respect to the first comment of the reviewer: we have extended our ChIP-qPCR analysis of OCT4, P300 and MED1 binding to additional sites (see new Supplementary Figure 8b) and precisely mapped the location of qPCR primers used on a detailed view of *Klf4*-associated SE locus showing ChIP-seq tracks for OCT4, P300, MED1 and ESRRB peaks (see new Supplementary Figure 8a). Regions that are either bound (DM1, DM2a and DM2b) or not bound (INT1, INT2 and C) by ESRRB were examined. Despite the relatively limited enrichment over background, our results validate the preferential binding of

proteins at DM versus INT regions in wild-type ESCs, especially at DM2 locations, and furthermore reveal the significant loss of MED1 binding in *Esrrb* ^{-/-} ESCs as consistently confirmed in an additional CHIP-qPCR experiment (now n=4). In contrast, we find that OCT4 and P300 binding are overall retained upon ESRRB depletion, as previously reported (Xie *et al.* Genes & Dev. 2018). Additionally, we provide novel evidence that reduced MED1 occupancy is accompanied by a loss of RNA polymerase II recruitment and eRNA detection along the *Klf4*-associated SE (see new Figure 6c,d and new Supplementary Figure 8d-f). Given the importance of MED1 in mediating promoter-enhancer interactions, we demonstrate that chromatin interactions at the *Klf4* locus are destabilised upon ESRRB depletion (see new Figure 6b).

- As suggested, we have examined the relationship between ESRRB occupancy and MED1 loss in the context of OCT4 binding using a fitted regression model (see text line 343, new method section line 704 and new Supplementary Table 9). No significant association between OCT4 binding score in wild-type ESCs and the loss of MED1 upon ESRRB depletion was found (Student's T test, p-value = 0.27), highlighting the specificity of ESRRB-MED1 association in our model system.

9. The loss of mediator provides an ad hoc explanation of the ESRRB-induced expression loss but has little to do with DNA methylation and the metastable state proposed by the authors. This reviewer recommends the authors to produce methylation data in ESRRB KO ESCs in order to link changes in ESRRB and methylation at SEs.

- As recommended by the reviewer, we have examined the impact of ESRRB depletion on CpG methylation along *Klf4*-associated SE used in this study as a model locus (see new Figure 5g). We show that the loss of ESRRB binding promotes *de novo* methylation in *Esrrb* ^{-/-} ESCs (see text line 309). Furthermore, we show that ESRRB is associated with DM subregions, which display higher CpG methylation variance at the single-cell level (see answer to reviewer 1, point 1)(see new Figure 4b and Supplementary Figure 6a). Finally, upon depletion of ESRRB, active marks such RNA polymerase II and eRNA expression are lost on the *Klf4* locus (see new Figure 6c,d and new Supplementary Figure 8e,f). Together, these mechanisms explain the reduced gene expression that we observed in ESRRB-depleted ESCs as well as EpiSCs that do not express ESRBB (Figure 5e and Supplementary Figure 7h).

Minor comments

1. To determine how SEs subregions interacts with promoters, the authors used a capture Hi-C dataset based on HindIII. This enzyme is a 6 bp cutter, and thereby cuts in average every 4kb, it is thus difficult to imagine how they could achieve a coverage good enough to make the difference between methylated and unmethylated subregions. Could the author comment on that?

- Our capture Hi-C data (Joshi *et al*; DpnII) and the dataset of Sahlen *et al.* (NcoI) are actually based on 4bp cutters (see text line 98) that cut DNA every 256 bp on average. PU and DM subregions have a median size of 900bp and are thus covered by 3 to 4 restriction fragments. Importantly, our capture Hi-C dataset enriches for all DNase1 hypersensitive sites in the genome (~100,000 baits; see Joshi *et al.* Cell Stem Cell 2015) that frequently overlap PU and DM subregions causing a further enrichment at these regions. The number of baited restriction fragments per subregions has a significant influence on the observed interaction frequency and is therefore typically higher at PU/DM regions. To this end, we

1) took the number of baits per subregion into account in our statistical model and 2) compared our data with the Sahlen *et al.* capture Hi-C dataset that enriches for promoter interactions only. Both analyses provided similar results, with an expected higher significance in our dataset.

Finally, the reviewer is correct that coverage at very small subregion is difficult to estimate. Therefore, we further omitted subregions smaller than 500bp from the analysis, as these are covered by 1 or 2 restriction fragments that often capture no or very few capture Hi-C interaction reads. We have now applied the proposed filtering to the new Figure 1d and new Supplementary Figure 3h,i in our revised manuscript, and highlighted this change in the Methods section line 647.

2. The authors use the term “accessible” to define unmethylated regions (see for instance line 116). This reviewer is not sure if this is an accepted terminology.

➤ We agree with the reviewer and this has now been amended by using the word “hypomethylated” instead of “accessible” as shown line 116 and 205.

3. The authors should mention in the text how many SEs are studied in this work and how many regions are classified as PU, DM or INT.

➤ This information is now included in the legend text of Figure 2.

4. It is unclear to this reviewer why the enrichment of H3K27ac is different between figures 3e,f and 4c-5e.

➤ Figure 3e,f (now Supplementary Figure 5d,e in the revised manuscript) show the relative enrichment levels of H3K27ac at PU, DM and INT subregions in EpiLCS and EpiSCs respectively, whereas Figure 4c and Figure 5e (now Figure 4d and Figure 5c, respectively) show the same features in ESCs grown in serum/LIF. For direct comparison and clarification, we have now included a new Figure 3e showing the relative enrichment of H3K27ac at PU, DM and INT in these three different cell populations in one single figure panel.

5. Why is the AF-2 mutant ESRRB affecting both class I and II genes set? This reviewer missed a discussion about this aspect.

➤ Here, we observed that overexpressing AF-2 mutant ESRRB leads to a loss of ESC self-renewal, and downregulation of pluripotency genes including Class I and Class II SE-associated genes. This suggests that the ectopic expression of AF-2 mutant ESRRB protein might destabilise the formation of activation protein complexes at SEs (as discussed in the text line 301).

Reviewer #3 (Remarks to the Author):

Bell et al. have identified that SEs can be subdivided in different regions with different methylation and TF binding properties. Interestingly, some of these regions are variably methylated and have high levels of binding of ESRRB while constant low methylated regions seem to be the main target regions for classical pluripotency factors. Overall this is a well performed study with interesting novel observations. My main concern is, that the observed differences attributed to methylation are a

consequence of CpG density and less of methylation itself. This should be better discussed in the text.

- We have added in our revised manuscript density plots showing how DM, PU and INT subregions are distributed in terms of CpG density (i.e. the number of CpG dinucleotides within a region divided by its length; see new Supplementary Figure 6e). As defined in Brinkman *et al.* Genome Res 2012, only regions with density above 0.05 would escape methylation. In contrast, regions harbouring density ranging from 0.03 to 0.05 and below 0.03 would be moderately and densely methylated, respectively. Based on this criteria, most DM (60%) and INT (80%) subregions would be predicted to be hypermethylated, while a large fraction (85%) of PU would show moderate to high methylation. In line with this prediction, we report that PU subregions acquire methylation in haematopoietic cells though at lower levels than DM and INT regions (see new Supplementary Figure 4g). Therefore, we agree with the reviewer that CpG density might contribute to shape the CpG methylation profiles of PU, DM and INT subregions to some extent. However, in contrast to somatic cells, PU, DM and INT regions show very different methylation profiles in pluripotent cells. This suggests that SE subregions are regulated by CpG methylation in a tissue-dependent manner, with PU subregions being transiently protected from *de novo* methylation in the developing epiblast prior to lineage specification as now discussed in text line 394. Please note that no (or minimum) overlap with CpG islands described as being resistant to CpG methylation is observed across all SE subregions (PU and DM 6%, and INT 8%, data not shown).

Comments:

Figure 1a) I think the representation of mCpG values as -1 to 1 a bit confusion. Could this not be shown as 0 – 1 which would be the normal expected range?

- In this study, we used the Hidden Markov model (HMM) to determine the methylation status (methylated/unmethylated) of each CpG (see Methods line 549). In order to include this information into a visual representation of CpG methylation on the browser, we chose to use a -1/1 scale, rather than % of methylation at each CpG position alone. Here, the height of each bar (each CpG) represents the exact % of methylation on a 0-1 scale for CpG determined as methylated, and the exact % of methylation on a -1-0 scale for CpG determined as unmethylated. Hence, a completely demethylated CpG will be -1. We agree with the reviewer that this is not a classic representation; however, it is a visual way to keep all information on CpG methylation at each position.

Is the difference in CpG density significant – it looks very minor and may be consequence of the relatively small size of SEs?

- We surmise that the reviewer is here referring to Supplementary figure 1. This figure has been edited and is now clearer. We chose to focus on ESC SEs, ESC TEs, and proB cell SEs, for which we had BS-seq data to compare (see below). CpG density is calculated as the number of CpG dinucleotide within a region divided by the length of this region and thus takes into account size differences between SEs and TEs. CpG density in the mouse genome is ~ 0.0083 . ESC TEs have a median value nearly identical to the genome value, while ESC SEs show a median value of ~ 0.012 . The difference is small but is statistically significant (Wilcoxon rank-sum test, p-values $< 2 \times 10^{-16}$).

Figure S1C: it would be nice to show the 5mC levels at proB SE and ESC SE in both conditions, ie proB SE in proB and also ESC

- We have now included the methylation profiles of ESC SEs and proB cell SEs alongside ESC TEs in B cells for which BS-seq data are available (see new Supplementary Figure 1c and text line 86).

Figure 1B: Many TFs are known to bind CpG rich motifs and in 1A the authors show that SE have mostly unmethylated CpGs -> is the result in 1B not just a consequence of this, ie are methylation levels relevant or just CpG position?

- This is an interesting suggestion; however, we note that not all TFs bound to unmethylated regions within SEs carry a CpG in their binding motifs. For instance, as shown in Figure 5b, the motifs of ESRRB, OCT4 and SOX2 binding sites are devoid of CpG. We do not agree with the reviewer that SEs have mostly unmethylated CpG. In contrary, we demonstrate that SEs partition into methylated (INT) and unmethylated (PU and DM) regions in ESCs grown under serum/LIF conditions, as previously suggested in other cellular contexts (Heyn *et al.* Genome Biol. 2016).

Figure 1C: see above – is it a consequence of methylation levels?

- As shown in Figure 1a and Figure 1b, unmethylated segments of SEs coincide with foci of protein binding and are preferred contact-points for SE-promoter interactions as opposed to methylated segments (Figure 1c and Supplementary Figure 2 for additional examples). Interestingly, we find that the prevalence of Hi-C chromatin interactions at unmethylated over methylated regions is conserved in ESCs under 2i/LIF conditions (see Supplementary Figure 3h,i), which are known to enforce a globally hypomethylated state of the genome (Leitch *et al.* Nature Structural & Molecular Biology 2013; Habibi *et al.* Cell Stem Cell 2013). This suggests that the structural organisation of SEs is primarily imposed by cell-type specific TFs, most likely counteracting CpG methylation at binding sites under permissive conditions as discussed in the text line 372. To further explore this, we have now tested the impact of ESRRB depletion on CpG methylation (new Figure 5g), and chromatin interactions at the *klf4* locus (see new Figure 6b). We show that promoter-SE interactions are significantly destabilised at this locus upon loss of ESRRB-MED1 occupancy, concomitant with the acquisition of CpG methylation in *Esrrb* -/- ESCs.

Line 116 “ESC SE-interacting gene promoters remained largely accessible” – this should say “largely hypomethylated”

- We would like to thank the reviewer for notifying this mistake. We have corrected it in our manuscript.

Figure 2,3: Interesting and convincing observation

- We thank the reviewer for this comment.

Figure 3e.f: This result seems expected based on the results shown before assuming that methylation levels are related to the factors measured here. The authors could briefly discuss/mention this.

- Figures 3e,f (now Supplementary Figures 5d,e in the revised manuscript) show heatmaps examining at PU (unmethylated), DM and INT (methylated) subregions the relative enrichment binding of general (OCT4 and SOX2) and primed (OTX2 and ZIC2) pluripotency TFs in EpiLCs and EpiSCs where these factors are expressed, validating the active status of PU subregions in the primed cells. To facilitate comparisons between the CpG methylation status of SE subregions, dynamics of TF binding and changes in chromatin accessibility in the transition towards primed pluripotency, we have now added a new figure panel (Figure 3e), showing a gradual decline of OCT4 binding and enhancer signatures (H3K37ac and ATAC-seq signals) as DM subregions become targeted by *de novo* methylation (see Figure 2c). In contrast, OCT4 remain bound to PU subregions, which appear constitutively accessible and unmethylated both in ESCs and EpiSCs.

Figure 4: Could the authors have predicted the PU and DM regions by looking at DNMT3a/b and TET1 binding profiles, ie how well would these overlap?

- To follow up on the suggestion of the reviewer, we used logistic regression models to test whether the status of PU and DM subregions can be predicted based on epigenetic features including DNMT3A/B and TET1 binding, enrichment levels for H3K27ac, H3K4me1 and H3K4me3, ATAC-seq signals and CpG density (see Methods line 694 and new Supplementary Table 8). Results show that all features tested are significantly associated with PU/DM status apart from H3K27ac deposition. As anticipated, we find that DNMT3 binding and H3K4me1 most closely associate with the DM status. In contrast, TET1, H3K4me3, and ATAC-seq signals are better predictors of the PU status. CpG density is also identified as a close predictor of PU subregions, in coherence with TET1 preferential binding to CpG-rich regions (Jin *et al.*, Nucleic Acids Res. 2014; Williams *et al.*, Nature 2011; Wu *et al.* Nature 2011). Only 17% of PU subregions, however, show CpG density scores high enough (above 0.05) to account for their unmethylated status as a consequence of CpG density alone (see text line 220).

Figure 5C: Does ESRRB/NR5A2 show a strong variable expression in serum/LIF cells, ie does ESRRB/NR5A2 expression correlate with the status of DM in single cell datasets?

- *Esrrb* and *Nr5a2* are both variably expressed in ESCs grown in serum/LIF (see new Supplementary Figure 6f). However, only *Esrrb* shows high predictive power to discriminate “naïve-like” from “primed-like” ESC sub-clusters, which are defined based on the methylation status of DM subregions at the single-cell level (see ROC analysis in new Figure 4f now including both *Esrrb* and *Nr5a2*)(see also text line 255).

Figure 7: I am not sure if the model should suggest a different binding kinetic for each SE region or whether the differences are more between cells. This is something that the authors could discuss.

- The proposed model builds upon our observations that 1) PU and DM are distinctively sensitive to the heterogeneous expression and binding of ESRRB in ESCs, and 2) the independent fates of PU and DM subregions upon exit of naïve pluripotency. This model does not infer any difference in the binding kinetic of ESRRB for each SE region. We have now edited the legend of Figure 7 to clarify this point.

Reviewers' comments:

Reviewer #1 (Remarks to the Author):

This revised version of the manuscript has significantly improved and the authors have addressed most of the concerns raised by the reviewers. Bell et al. now characterise in greater detail the effects of loss of *Esrrb* on DNA methylation at super enhancer sub-regions, mediator recruitment, marks of activity and the establishment of topological contacts with target promoters. The discussion of results from the analysis of genome-wide methylation profiling in naïve and primed cells is more clear, and conclusions better linked to the existing literature. Overall, this work now provides solid data describing super-enhancers as a collection of individual, and often independent, subunits and explores the role of *Esrrb* in controlling enhancer function in pluripotent cells.

Suggestions:

A) In order to strengthen the link between loss of *Esrrb* binding and methylation changes at DM regions, it would be helpful to compare *de novo* methyltransferase occupancy at selected enhancer regions in WT and *Esrrb* null ES cells.

B) Most of the analysis of the effects of *Esrrb* depletion is performed comparing WT and *Esrrb* null ES cells cultured in FCS/LIF conditions. As the authors acknowledge in their previous version of the manuscript, and as suggested by their gene expression analysis, in these conditions *Esrrb* null ES cells show extensive spontaneous differentiation. This opens the possibility that the observed changes in methylation, marks of activity and topological organization at enhancer regions are partially indirect and due to the presence of a fraction of differentiated cells in the cultures. Since the self-renewal of *Esrrb* null ES cells is stabilised in 2i/LIF, repeating a minimal selection of the experiments presented comparing WT and KO cells in such culture conditions would significantly strengthen the conclusions of this work. For instance, DNA methylation, mediator occupancy and eRNA levels could be analysed at a few selected super-enhancer sub-regions.

Additional points:

In support of their results, the authors could discuss the notion that *de-novo* DNA methyltransferases, although not essential, are important to allow a timely exit from naïve pluripotency (Li et al., A lncRNA fine tunes the dynamics of a cell state transition involving *Lin28*, *let-7* and *de novo* DNA methylation, *eLIFE*, 2017). In addition, the finding that *Dnmt3a* is specifically acting on pluripotency enhancers during differentiation is relevant to the results presented in this study (Petell et al., An epigenetic switch regulates *de novo* DNA methylation at a subset of pluripotency gene enhancers during embryonic stem cell differentiation, *NAR*, 2016), as is the report that the *de novo* methyltransferases *Dnmt3a/b* but not Tet enzymes affect methylation levels at *Sox2* super-enhancer (Song et al., Dynamic Enhancer DNA Methylation as Basis for Transcriptional and Cellular Heterogeneity of ESCs, *Mol Cell*, 2019).

The authors could consider acknowledging that the link between *Esrrb* downregulation during ESC differentiation and methylation changes at pluripotency enhancers has been previously described (Festuccia et al., *Esrrb* extinction triggers dismantling of naïve pluripotency and marks commitment to differentiation, the *EMBO journal*, 2018).

It would be helpful to include in the methods section a brief description of the analysis performed to generate Figure 4e. Based on the text included in the manuscript, it is at the moment difficult to understand if the plot presents the distribution of the variance in CpG methylation levels among all different DM sub-regions for each individual cell, or the variance in CpG methylation levels for single DM sub-regions (for which sequencing coverage is sufficient) across all cells analysed. Either approach would be differently informative.

Nicola Festuccia

Reviewer #2 (Remarks to the Author):

I found that the manuscript has substantially improved from the last version both in clarity and in content. The authors have satisfactorily answered all my previous points. However, I still have two minor comments for the authors to answer.

Minor comment:

1) Fig 6b and line 330: the interaction between Klf4 and its SE is mostly changing between cells that are cultured in 2i or serum/LIF. In comparison to this, the loss of interaction in *Esrrb*^{-/-} cells is marginal. If the authors want to link the loss of interaction with loss of MED1 binding, I would suggest them to display a 2i-cultured MED1 ChIP-seq track to compare with the serum/LIF cells. If not, I would recommend the authors to tone down their statement on MED1-mediated interaction.

2) Fig 6a: in the p300 ChIP, it seems that the averages (represented by the bar summit) do not fit the average from the various datapoints (dots).

Reviewer #3 (Remarks to the Author):

The revisions made by Bell and colleagues well address the raised points. This is a nice and comprehensive analysis of SE "types" and how these different types may have a functional role. My prior concerns regarding the big role of CpG density have been addressed as well as other points raised and I do not have further comments to add.

Responses to reviewers_ revision 2_NCOMMS-19-08409

Reviewers' comments:

Reviewer #1 (Remarks to the Author):
Nicola Festuccia

This revised version of the manuscript has significantly improved and the authors have addressed most of the concerns raised by the reviewers. Bell et al. now characterise in greater detail the effects of loss of *Esrrb* on DNA methylation at super enhancer sub-regions, mediator recruitment, marks of activity and the establishment of topological contacts with target promoters. The discussion of results from the analysis of genome-wide methylation profiling in naïve and primed cells is more clear, and conclusions better linked to the existing literature. Overall, this work now provides solid data describing super-enhancers as a collection of individual, and often independent, subunits and explores the role of *Esrrb* in controlling enhancer function in pluripotent cells.

We thank the reviewer for his enthusiasm about our study and positive evaluation of our revised manuscript.

Suggestions:

A) In order to strengthen the link between loss of *Esrrb* binding and methylation changes at DM regions, it would be helpful to compare *de novo* methyltransferase occupancy at selected enhancer regions in WT and *Esrrb* null ES cells.

Given the low enrichment and broad genomic binding patterns of DNMT3A and DNMT3B in ESCs (Baulec et al. Nature 2015) along with a lack of *Dnmt3a/b* induction in *Esrrb*^{-/-} ESCs (see new Supplementary Figure 7h), it is unlikely that examining the occupancy of DNMT3s at the *Klf4*-associated SE (or any other SEs) in control and *Esrrb*^{-/-} ESCs would be resolute enough to reliably detect changes in DNMT3s binding across individual SE subregions. Hence, we chose during our previous revision work to monitor the enzymatic activity of DNMT3s as a good proxy for binding by quantitatively measuring sites of *de novo* methylation within DM and INT subregions of the *Klf4* SE (see Figure 5g and method section lines 523-532).

Data showed an increase in CpG methylation at all DM CpG sites examined in *Esrrb*^{-/-} ESCs (see Figure 5g). This might either be due to increased DNMT3A/DNMT3B binding specifically at DM subregions or spreading of CpG methylation from DNMT3B highly-bound, interstitial (INT) segments, as we find that these regions are not affected by the loss of *ESRRB* binding. As now discussed in our revised manuscript lines 428-430, methylation spreading from INT to DM subregions could indeed be facilitated by the processive nature of DNMT3B upon protein binding destabilisation as previously suggested (see Gowher and Jeltsch, J. Biol. Chem. 2002, Baulec et al. Nature 2015, Norvill et al. Biochemistry 2018 amongst other studies).

It is interesting to note that while INT regions are highly enriched for DNMT3B, DNMT3A together with TET molecules might more closely delineate DM subregions (see Figure 4d) as now further highlighted in our revised manuscript lines 228-229. Further exploring the specific actions of DNMT3A and DNMT3B at SE subregions, the precise relationship between *ESRRB* and DNMT3A/B as well as the molecular mechanisms governing the metastability of DM subregions in ESCs (i.e. targeted recruitment of DNMT3s versus CpG methylation spreading upon loss of *ESRRB* binding) are all very exciting questions arising from our novel findings and previous studies. Addressing these questions, however, is beyond the scope of this paper and thus are best suited for follow-up investigations.

B) Most of the analysis of the effects of *Esrrb* depletion is performed comparing WT and *Esrrb* null ES cells cultured in FCS/LIF conditions. As the authors acknowledge in their previous version of the manuscript, and as suggested by their gene expression analysis, in these conditions *Esrrb* null ES cells show extensive spontaneous differentiation. This opens the possibility that the observed changes in methylation, marks of activity and topological organization at enhancer regions are partially indirect and due to the presence of a fraction of differentiated cells in the cultures.

We disagree with the reviewer that ESCs constitutively depleted for ESRRB (in contrast to conditionally depleted cells) show extensive spontaneous differentiation in serum/LIF conditions. As shown by Martello *et al.* originally reporting the phenotype of the *Esrrb*^{-/-} ESCs used in this study, these cells maintain an undifferentiated state upon serial passages in serum/LIF showing no induction of primed differentiation markers (e.g. *Otx2*, *Fgf5*, *Dnmt3b*) and retained the expression of *Pou5f1*, *Sox2* and *Nanog* (amongst other pluripotency markers) (Martello *et al.* Cell Stem Cell 2012; see also Supplementary Figure 7h). This point has now been clarified in our revised manuscript lines 319-322.

In contrast, we report that ectopically expressing an AF-2 mutant form (ESRRB MutAF-2) as opposed to wild-type form (ESRRB WT) in *Esrrb*^{-/-} ESCs massively triggers spontaneous differentiation. Thus, ESRRB MutAF-2 cells cannot be propagated in culture as acknowledged in our original manuscript, and further clarified in our revised manuscript lines 313-314.

Since the self-renewal of *Esrrb* null ES cells is stabilised in 2i/LIF, repeating a minimal selection of the experiments presented comparing WT and KO cells in such culture conditions would significantly strengthen the conclusions of this work. For instance, DNA methylation, mediator occupancy and eRNA levels could be analysed at a few selected super-enhancer sub-regions.

We also disagree with the statement of the reviewer that *Esrrb*^{-/-} ESCs can be stabilised in 2i/LIF. In our hands, we find that these cells die when propagated under 2i/LIF conditions (as also reported upon acute deletion of ESRRB; see Atlasi *et al.* Nature Cell Biology 2019), precluding the possibility to perform the suggested additional experiments. This information has now been included in our revised manuscript lines 1073-1074 (see legend text of Figure 5g).

Additional points:

In support of their results, the authors could discuss the notion that de-novo DNA methyltransferases, although not essential, are important to allow a timely exit from naïve pluripotency (Li *et al.*, A lncRNA fine tunes the dynamics of a cell state transition involving Lin28, let-7 and de novo DNA methylation, eLIFE, 2017).

As kindly suggested by the reviewer, this paper is now cited in our revised manuscript in support of our model depicted in Figure 7 – see revised text lines 454-455.

In addition, the finding that Dnmt3a is specifically acting on pluripotency enhancers during differentiation is relevant to the results presented in this study (Petell *et al.*, An epigenetic switch regulates de novo DNA methylation at a subset of pluripotency gene enhancers during embryonic stem cell differentiation, NAR, 2016), as is the report that the de novo methyltransferases Dnmt3a/b but not Tet enzymes affect methylation levels at Sox2 super-enhancer (Song *et al.*, Dynamic Enhancer DNA Methylation as Basis for Transcriptional and Cellular Heterogeneity of ESCs, Mol Cell, 2019).

These two relevant papers have also been included in our revised text lines 228-229 (Petell *et al.* 2016), and both line 198 (as previously) and 250-251 (Song *et al.* 2019) as suggested.

The authors could consider acknowledging that the link between *Esrrb* downregulation during ESC differentiation and methylation changes at pluripotency enhancers has been previously described (Festuccia *et al.*, *Esrrb* extinction triggers dismantling of naïve pluripotency and marks commitment to differentiation, the EMBO journal, 2018).

This paper was already cited in our manuscript line 327 and line 425, and now additionally quoted lines 444-447 reporting the acquisition of CpG methylation at the *Esrrb*-associated SE in cells initiating differentiation (i.e. sorted *Esrrb*-GFP negative ESC subpopulations). This is in agreement with our computational BS-seq analyses performed at the single-cell level looking at the *Esrrb* SE and other SEs mapped in ESCs (see Figure 4a-c and Supplementary Figure 6b).

It would be helpful to include in the methods section a brief description of the analysis performed to generate Figure 4e. Based on the text included in the manuscript, it is at the moment difficult to understand if the plot presents the distribution of the variance in CpG methylation levels among all different DM sub-regions for each individual cell, or the variance in CpG methylation levels for single DM sub-regions (for which sequencing coverage is sufficient) across all cells analysed. Either approach would be differently informative.

Additional information has been included in our revised Method section lines 700-704, clarifying that the variance in CpG methylation levels shown in Figure 4e and Supplementary Figure 6a was computed among all different subregions for each individual cell analysed in a given treatment condition (e.g. DNMT3s DKO and TETs TKO in Figure 4e).

Reviewer #2 (Remarks to the Author):

I found that the manuscript has substantially improved from the last version both in clarity and in content. The authors have satisfactorily answered all my previous points. However, I still have two minor comments for the authors to answer.

Minor comment:

1) Fig 6b and line 330: the interaction between *Klf4* and its SE is mostly changing between cells that are cultured in 2i or serum/LIF. In comparison to this, the loss of interaction in *Esrrb*^{-/-} cells is marginal. If the authors want to link the loss of interaction with loss of MED1 binding, I would suggest them to display a 2i-cultured MED1 ChIP-seq track to compare with the serum/LIF cells. If not, I would recommend the authors to tone down their statement on MED1-mediated interaction.

We have now added MED1 ChIP-seq track collected from ESCs grown in 2i/LIF to compare with the serum/LIF cells (see new Figure 6b panel and revised text lines 342-343). Mirroring the stronger and weaker 4C-seq interactions detected in 2i/LIF and serum/LIF ESCs respectively, we show that MED1 occupancy is higher in 2i/LIF relative to serum/LIF cells at the *Klf4* locus, in further support of a role for MED1 in regulating chromatin interactions.

Additionally, we examined and plotted 4C-seq reads across *Klf4*-associated SE subregions in control and *Esrrb*^{-/-} ESCs grown in serum/LIF conditions, demonstrating a significant loss of interactions in the absence of ESRRB (see new Supplementary Figure 8c and revised text lines 346-348), especially at DM subregions where most promoter-SE interactions assemble (see also Figure 1c).

2) Fig 6a: in the p300 ChIP, it seems that the averages (represented by the bar summit) do not fit the average from the various datapoints (dots).

We thank the reviewer for pointing this out; this has now been corrected.

Reviewer #3 (Remarks to the Author):

The revisions made by Bell and colleagues well address the raised points. This is a nice and comprehensive analysis of SE “types” and how these different types may have a functional role. My prior concerns regarding the big role of CpG density have been addressed as well as other points raised and I do not have further comments to add.

We thank the reviewer for their complements on our “nice and comprehensive analysis of SE types”. This reviewer has had no further comment to report.